



# Seasonal evolution of a ski slope under natural and artificial snow: detailed observations and modelling

Pierre Spandre[1,2], Hugues François[1], Emmanuel Thibert[1], Samuel Morin[2], and Emmanuelle George-Marcelpoil[1]

[1]Université Grenoble Alpes, Irstea, Grenoble
[2]Météo-France CNRS, CNRM-GAME UMR 3589, Centre d'Etudes de la Neige, Grenoble

*Correspondence to:* Pierre Spandre (pierre.spandre@irstea.fr)

**Abstract.** The production of Machine Made (MM) snow is now generalized in ski resorts and represents the most common adaptation method to mitigate the impacts of both the natural variability and projected changes of the climate on the snow conditions to guarantee suitable conditions for skiing. Most investigations of the impact of snow conditions on the economy of the ski industry under past, present or projected climate focus on the production of MM snow. So far, none of them accounted

for the efficiency of the snowmaking process i.e. the actual MM snow mass that can be recovered from a given water mass used for snowmaking. The present study consisted in observations and interpolation on a 0.25 $m^2$ grid of snow conditions (depth and mass) using a Differential GPS method and snow density coring, after single sessions of production (prior to MM snow spreading by grooming machines) and on the ski slope as opened to skiers, on a beginner trail in Les Deux Alpes ski resort (French Alps). A detailed physically based snowpack model accounting for grooming and snowmaking was used to address the

seasonal evolution of the snowpack and compared to the observations. Our results show that approximately 30% of the water mass can be recovered as MM snow within 10 m from the center of a MM snow pile after the production and 50% within 20 m. The observations and simulations on the ski slope were relatively consistent with 60% ($\pm$ 10%) of the water mass used for snowmaking within the edge of the ski slope. We also addressed the losses due to thermodynamic effects resulting in less than 10% of the total water mass in the present case. The main uncertainty pertains to the surface of observations: the surface of the

ski slope opened to skiers changed along the season and objective uncertainties exist, in particular from man-made decisions. These results suggest that even in the ideal conditions for production a significant fraction of the water used for snowmaking can not be found as MM snow within the edge of the ski slope with most of the lost fraction of water due to site dependent characteristics (e.g. meteorological conditions, topography, human decisions).

## 1 Introduction

Snow is a vital material for the ski industry (Fauve et al., 2002) encouraging ski lifts operators to increasing technical methods of snow management to mitigate their dependency to the variability of both the quality (Armstrong and Brun, 2008) and the quantity of snow (Durand et al., 2009; Hughes and Robinson, 1996). Thanks to grooming and snowmaking ski resorts operators intend to provide steadier and safer conditions (Bergstrom and Ekeland, 2004) and to ensure the operation of ski





facilities (Hopkins, 2013; Trawöger, 2014). Ski resorts stakeholders also rely on snowmaking to mitigate the effects of climate change on the snow conditions (Hopkins and Maclean, 2014; Morrison and Pickering, 2012). Snowmaking has therefore been the main concern of recent investigations of the impacts of climate change on the ski industry (Scott et al., 2003; Hennessy et al., 2007; Steiger, 2010; Pütz et al., 2011; Damm et al., 2014). To the best of our knowledge though, none of these results

accounted for the efficiency of the snowmaking process i.e. the actual conversion of water volumes used for production into Machine Made snow on ski slopes (MM, Fierz et al. (2009)) while the related water losses may be significant (Eisel et al., 1990; Spandre et al., 2016) and of great interest for operational purposes (technical issues, investments).

The water losses during snowmaking were addressed in a few studies with different approaches (modelling, interviews, observations) and investigated factors (thermodynamic processes, mechanical wind erosion, etc.). Eisel et al. (1988) estimated

the consumptive water loss during the snowmaking process through evaporation and sublimation by a combination of nine field experiments (mass balance) and a theoretical approach (energy balance). They found an average 6% water loss and a negative linear relationship between the atmospheric temperature and the water loss. Hanzer et al. (2014) implemented the relationship derived by Eisel et al. (1988) in a detailed snowpack model and found that for typical snowmaking conditions, water losses due to evaporation and sublimation ranged between 2 to 13%. Although the experiments by Eisel et al. (1988) remain the

most detailed at this date, they were not performed in operational conditions (low water flows, a maximum 4 m$^3$ of water used for production) and with a Machine Made snow (MM, Fierz et al. (2009)) technology which is outdated today. Eisel et al. (1990) later showed that water losses during snowmaking could not be satisfyingly limited to evaporation and sublimation alone by comparing the runoffs simulated by a hydrological model with the observations in six test sites in Colorado ski areas. An additional 7 to 33% loss was deduced after the initial loss (related to evaporation and sublimation), resulting in a total

consumptive loss of 13 to 37% range. A detailed investigation of the potential sources of water losses could not be computed due to the large scale of the experiments (overall basin catchment). Recently, Olefs et al. (2010) reported from interviews with professionals that water losses due to evaporation, sublimation and wind erosion were estimated between 15 to 40 % for air-water guns and 5 to 15% for fan guns. Spandre et al. (2016) performed observations on four ski slopes and found a minimum water loss over 25% with significant differences between sites of observations (some exceeding 50%) and concluded

that external factors (wind, topography, vegetation) had probably significant impacts on the efficiency of MM snow, although failing to provide accurate evaluations due to large uncertainties related to the low spatial resolution of observations.

The present study aims to provide a detailed description of the seasonal evolution of a ski slope snowpack in operational conditions with a high spatial resolution (0.5 x 0.5 m grid), including the additional MM snow by snowmaking facilities. The equivalent water masses of MM snow piles were assessed prior to any action by the grooming machines through dedicated

sessions, and both snow depth (SD) and snow water equivalent (SWE) of the prepared ski slope were observed in several occasions. These observations were crossed with all available data on the snow production (water flow, temperature, wind) and with the results of simulations using a detailed physically based snowpack model (Spandre et al., 2016) to compute the ratio of MM snow mass on the ski slope by snowmaking with respect to the water mass used for the production of MM snow (ratio defined as the Water Recovery Rate, WRR). The method is described in a first section, including all measurements and tests we



| First Natural snowfall | 21 Nov. | | | | | | |
|---|---|---|---|---|---|---|---|
| MM snow Obs. | | 23 Nov. | 24 Nov. | 28 Nov. | 1 Dec. | | 21 Jan. |
| Ski slope Obs. | | | | | 4 Dec. | 20 Jan. | 6 Apr. |
| Resort opening / closing | | | | | 5 Dec. | | 30 Apr. |
| Total melt-out | | | | | | | 3 May |

**Table 1.** Important dates of the field campaign carried out during the 2015-2016 winter season. "MM snow Obs." and "Ski slope Obs." correspond respectively to dates when observations were performed on MM snow piles and ski slope.

set up to characterize the uncertainties related to our measurements. From the results of these tests, the retained uncertainties and the results of observations are detailed in a second section and discussed.

## 2 Material and Methods

### 2.1 Description of observations: study area

5  The "Coolidge" ski slope is a beginner trail close to the village of Les 2 Alpes ski resort (Oisans range, French Alps) at an elevation of 1680 m.a.s.l. The area is mainly westward oriented and nearly flat (slope $\approx 5°$). This is an important slope within the resort which is used for skiing lessons and as a way back down to the village by ski, constraining technical services to keep it under operational conditions for skiing from the opening (early December) to the closing date of the resort (late April). Two distinct series of observations were carried out on this site during the 2015-2016 winter season (Table 1):

10  – Volume measurements of recently produced MM snow piles and the related mass. Five production sessions were observed (Table 1).

– Spatial measurements of snow depth (SD) and snow water equivalent (SWE) on the prepared ski slope i.e. in the skiing conditions as offered to skiers. Three observations were carried out (Table 1).





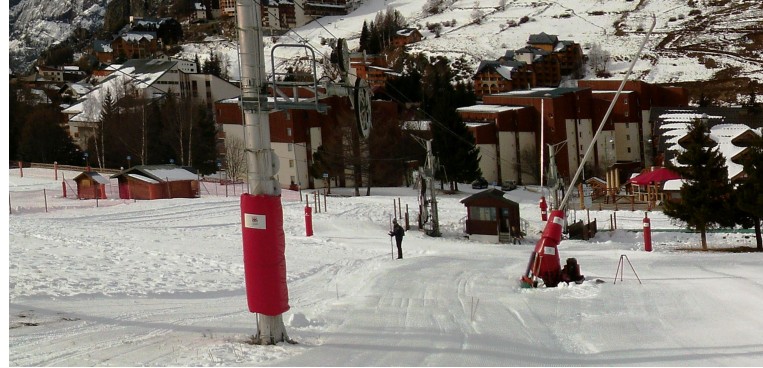

**Figure 1.** The Coolidge ski slope conditions on the 4 December 2015, the day before the resort opened. Edges with unprepared areas and obstacles (trees, lift infrastructures, snowgun) of the ski slope can clearly be seen.

The professional snowmakers of Les 2 Alpes kindly and openly provided all available data regarding the production of MM snow on the study site. This covers 15 min time step records of the water flow of the snowgun ($m^3\ h^{-1}$), the wet-bulb temperature (°C), the wind speed ($m\ s^{-1}$) and direction (° from North) measured in the vicinity of the study area. These data were used both as inputs to force the snowpack model (water flow, amount of MM snow) and as references to analyse the outputs of the model (same variables, wet-bulb temperature). The data also helped to characterize the production conditions (temperature and wind conditions).

The study area was defined from the local topography and the initial surface of the ski slope. The Coolidge ski slope is a large (up to 75 m large from January to March) and relatively flat grass covered area. In such case defining limits to the ski slope can be tricky and subjective. In order to be as objective as possible and consistent over the season we defined the following rules which were systematically applied:

- All MM snow piles were measured on the total surface where MM snow was observed, unless a major obstacle (tree, building) stood in the area, which we bypassed.

- The surface of the operational ski slope defined by the ski patrollers changed along the ski season by a factor of up to 1.75, depending on the snow conditions. The ski slope was wider in January (6632 $m^2$) and April (7067 $m^2$) than on the 4 December (4063 $m^2$) since there was very little natural snow at this time. This also made the edge easier to identify (Figure 1). On the 20 January and 6 April we collected data over the total marked out ski slope even though the study area for SD and SWE calculations was consistently limited to the area defined by the edge on the 4 December 2015 to provide comparable data. A sensitivity test of the SD and SWE to the study surface was conducted by considering an offset of ± 2 m of the edge. The impact on SD and SWE was computed and discussed.

- The surface considered to calculate the MM snow production rate in the model was defined as the total marked out ski slope area: the "useful" area (Figure 2). Beyond the initial MM snow production (late November) the natural snowfalls occured and the ski slope was enlarged. We expected the production to be spread over this area.





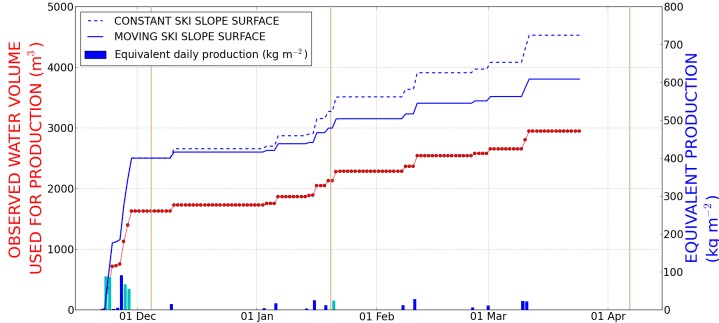

**Figure 2.** Observed water volumes used for production (red) and the equivalent mass on the ski slope surface (blue). Bars stand for the daily production (bottom). Production sessions when observations were performed on MM snow piles (cyan bars) and dates when ski slope observations were carried out (light brown) are outlined.

The relationship between the average snow depth (Section 2) and the study surface (defined by the 2015-12-04 edge) was tested by comparing the calculated snow depth within the study area ($4063 m^2$) and within bufferized surfaces of $\pm$ 2 m from the edge of the study area (respectively 3425 and 4749 $m^2$). The differences between bufferized snow depth and the snow depth calculated for the study area are consistent over the three observations sessions (data not shown). The larger the surface i.e.

the further the edge from the snowgun, the smaller the average snow depth. The average difference is + 0.03 m and - 0.03 m for respectively the smaller (- 2 m) and larger (+2 m) areas, and showed little variation from an observation session to antother (5% relative difference maximum), suggesting that the surroundings of the study area undergo consistent evolutions through the season and that the most important thing to address the evolution of the snowpack from the initial observation is to follow the exact same area. This also tends to confirm that the MM snow produced after the 4 December 2015 was actually spread

over the total useful surface after the slope was enlarged.

## 2.2 Snow depth measurement method and related uncertainties

### 2.2.1 Snow surface elevation point measurements

The snow surface elevation was measured in several occasions (Table 1) thanks to a geodetic double frequency Differential GNSS (GPS + GLONASS) Leica GS10 high precision receiver. A permanent frame was set up close to the study area on

17 November 2015 to provide a positioning antenna carrier at the reference station. The position of the GPS antenna once mounted on this frame was post processed to obtain the absolute position of the reference station within a few centimeters. To measure points coordinates in the investigated area, we used a rover receiver operating in real time kinematics (RTK) from the reference station. Specific points were defined (painted dots on concrete ground) and systematically re-measured during each GPS session as a control. The baseline (reference-to-rover) was less than 500 m for all sessions which ensures a relative position

from the reference station with a spatial (3D) accuracy below 0.02 m. The intrinsic uncertainty on the Z (altitudinal) position



of the Differential GPS was 0.012 m over all the observations sessions. The average density of points for the measurement of the elevation of the MM snow piles surface was 11.1 $m^2$ per point ($\pm$ 3.3 $m^2$ per point) i.e. each point covered a surface equivalent to a 1.88 m radius disk ($\pm$ 0.3 m). The average density for the measurement of the elevation of the ski slope surface was 16.4 $m^2$ per point ($\pm$ 4.4 $m^2$ per point) i.e. each point covered a surface equivalent to a 2.29 m radius disk ($\pm$ 0.31 m). The point density was adapted to the local conditions (terrain complexity), for each session i.e. the larger the changes of the snow surface, the more points were taken. This explains why the average surface per point for the measurement of the elevation of the ski slope surface (when snow surface is equalized by grooming machines) is larger than for MM snow piles sessions.

The bare ground surface elevation was also measured on the 17 November 2015 to be compared with the snow-free helicopter-borne laser scan Digital Elevation Model (DEM) of the area acquired in November 2015. Before checking the elevation consistency between our GPS survey and this snow-free DEM, we adjusted (-0.0032 m) the elevation of our reference station on a local (800 m apart) common levelling control point provided by Institut Géographique National (IGN).

### 2.2.2 Interpolation on a regular grid

In order to compare snow surfaces elevations with each other or with the DEM of the bare ground data need to be interpolated on a regular grid. The existing snow-free DEM had a spatial resolution of 0.5 m (0.25 $m^2$ pixels) which we decided to choose as the working grid. All data were interpolated on this grid thanks to a preliminar Triangular Irregular Network (TIN) method with a Delaunay natural neighbour triangulation (Maune, 2007). The same method was used to treat all sessions of observations. Once interpolated on the working grid, all observations sessions could be compared to each other or with the bare ground which provided a spatial observation of the snow depth over the study area.

Such a method carries several sources of uncertainty (instrument, interpolation) we intended to assess through three distinct tests:

- a high-resolution Terrestrial Laser Scan (TLS, Prokop (2008)) was used on the 1 December 2015 on a MM snow pile that we also measured with the GPS method. We could compare the differences of both the GPS points alone and the interpolated points with the TLS points to obtain the error carried out when interpolation is done (effect of point density).

- differences with the DEM of the bare ground of both the GPS points alone and the interpolated points of the bare ground by the GPS method (17 November 2015) were calculated

- hand made snow depth measurements were made on three occasions (observations of the ski slope) with a probe and compared with the interpolated snow depth by the GPS method.

### 2.2.3 Evaluation of related uncertainties on snow depth

We first compared the interpolated snow surface elevations with data from a Terrestrial Laser Scan. We used an Optech Ilris-LR laser scaner whose wavelenght (1064 nm) is adapted to the low reflectance of the snow in the infra-red spectrum. The laser scan point cloud was adjusted on targets whose coordinates were determined help to a total station. The internal consistency of the





target network was $\pm$ 0.0038 m and its relative positioning with respect to the GPS reference station was 0.008 m in planimetry and 0.013 m in elevation. An average - 0.012 m average elevation difference was measured between the GPS interpolated and the TLS snow surfaces (2018 m$^2$). The Root Mean Square of the differences (RMSD) was 0.055 m (Table 2). A significant variability (standard deviation) was measured within each 0.5 x 0.5 m$^2$ pixel thanks to the TLS measurements: 0.031 m on

average over the 8072 pixels. We conducted a Shapiro-Wilk test for normality (Royston, 1982) over the differences between interpolated elevations and the TLS measurements (see below). This suggests that the differences on snow surface elevation should not be considered as normally distributed even though the distribution looks very consistent with normality (Figure 2a).

- Statistical value w = 0.979

- p-value = 1.48 $10^{-25}$ ($<$ 0.05)

Second we compared the interpolated snow-free surface elevations from the existing Digital Elevation Model of the ground. An average 0.003 m average elevation difference was measured between the GPS interpolated ground surface and the DEM data (4044 m$^2$). The standard deviation of differences was 0.064 m (Table 2). We conducted again a Shapiro-Wilk normality test (Royston, 1982) over the differences between interpolated ground elevations and the Digital Elevation Model data (see below). This suggests that the differences should not be considered as normally distributed even though the distribution looks

very consistent with normality (Figure 2a).

- Statistical value w = 0.951

- p-value = 1.59 $10^{-34}$ ($<$ 0.05)

Last, the GPS interpolated snow depth was compared with hand made measurements in several occasions (Table 2, Figure 3). An average - 0.008 m average difference was measured between the GPS interpolated snow depth and the manual observations.

The standard deviation of differences was 0.053 m (Table 2). We conducted a Shapiro-Wilk normality test (Royston, 1982) over the differences between interpolated snow depth and the manual measurements (see below). This suggests that the differences on snow depth are normally distributed:

- Statistical value w = 0.963

- p-value = 0.38 ($>$0.05)



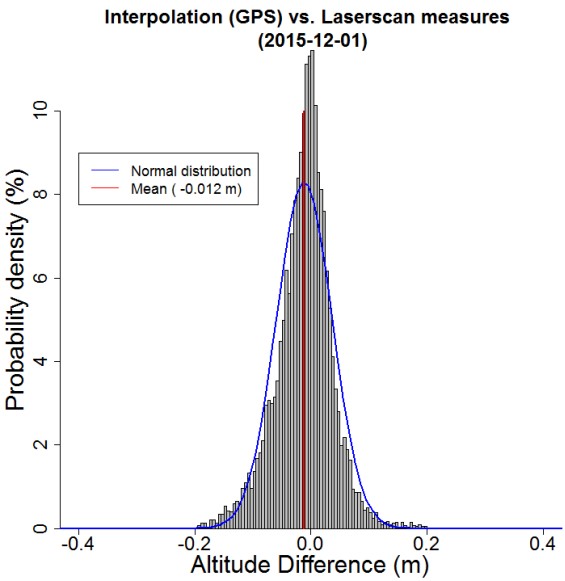

**Figure 2a.** Probability density of the elevation differences between the interpolated snow surface and the TLS snow surface on the 1 December 2015

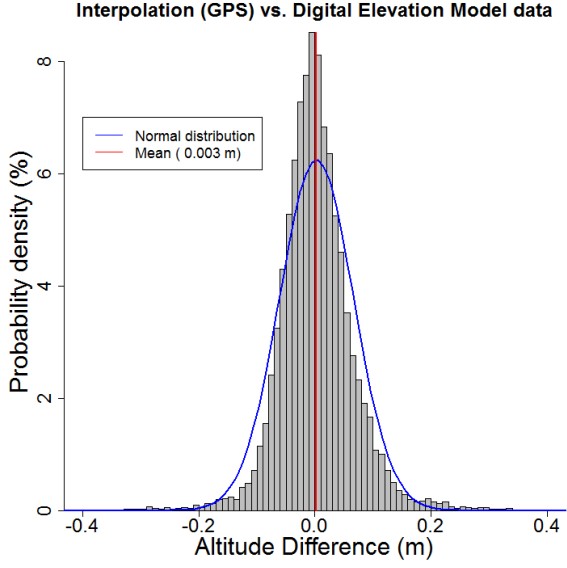

**Figure 2b.** Probability density of the elevation differences between the interpolated bare ground surface and the Digital Elevation Model ground surface





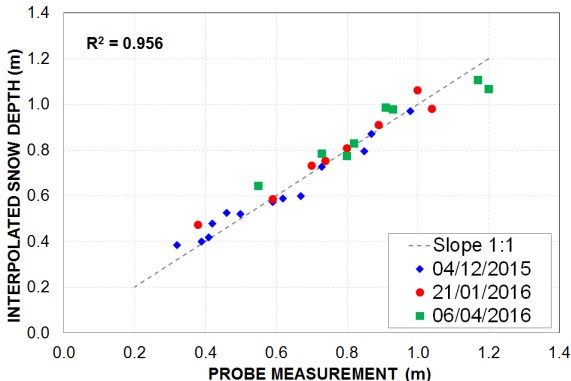

**Figure 3.** Interpolated snow depth from GPS method with respect to the hand made probe measurements for each observations session of ski slope. Average difference and RMS of the differences are detailed in Table 2

To conclude on the analysis of uncertainties, the average difference of the snow surface elevation interpolated from the Differential GPS points with respect to either TLS measurements or DEM data ranged between - 0.012 and 0.003 m (Tables 2) while the RMS of the differences ranged between 0.048 and 0.064 m. The distribution should statistically not be considered as normally distributed. However distributions are close in both cases to normality (Figure 2a and 2b). Beyond these results we compared the interpolated snow depth with hand made measurements. The agreement was excellent with an average error of - 0.008 m (RMSD = 0.053 m) and a statistically significant test for normality (Table 2, Figure 3). Regarding the internal variability of the snow surface elevation within a pixel (0.031 m) and the sensitivity of the snow depth to the study area, we therefore considered $\sigma_{SA}$ = 0.03 m as the uncertainty on the elevation. We also considered the error on the snow surface elevation to be normally distributed, which appeared as a reasonnable approximation; Consequently the combined uncertainties to obtain the snow depth uncertainty $\sigma_{SD}$ can be deduced (Bevington and Robinson, 2003) assuming errors to be uncorrelated, which provides a consistent value to the calculated RMS of the differences with the comparison methods (Table 2):

$$\sigma_{SD} = \sqrt{2} * \sigma_{SA} = 0.042 \text{m} \tag{1}$$




| Comparison method | Type / Session | Number of points | Average difference (m) | RMS of differences (m) |
|---|---|---|---|---|
| Terrestrial Laser Scan | GPS points | 156 | - 0.0046 | 0.055 |
| | Interpolated points | 8072 pixels (2018 m$^2$) | - 0.012 | 0.048 |
| Digital Elevation Model | GPS points | 145 | 0.032 | 0.047 |
| | Interpolated points | 16179 pixels (4044 m$^2$) | 0.003 | 0.064 |
| Probe manual measurements | 2015-12-04 | 13 | - 0.002 | 0.041 |
| | 2016-01-20 | 8 | - 0.019 | 0.046 |
| | 2016-04-06 | 8 | - 0.006 | 0.073 |
| | All | 29 | - 0.008 | 0.053 |

**Table 2.** Average difference and RMS of the differences between interpolated snow surface elevation and Terrestrial Laser Scan measurements on a snow pile (1 December 2015), between the elevation of the bare ground by the GPS method and the existing Digital Elevation Model (DEM) and between interpolated snow depths and probe measurements on ski slopes (Figure 3).

## 2.3 Conversion of snow volumes into snow masses

The snow density was measured when possible on MM snow piles by weighing 1/2 liter snow samples. We used the average density and standard deviation of all observations together for the sessions when we could not perform density measurements (23 November and 1 December 2015). We also performed measurements of the average density of the snowpack on the ski slope using a PICO coring auger (Koci and Kuivinen, 1984) for each session of observations. The density showed a rather weak variation from a production session to another, resulting in a relative error of 4% on MM snow density (Table 3). This supported the assumption of using the average and standard deviation of density over all observations regardless the dates when measurements were missing. The snowpack average density on the ski slope showed a significant increase in the season with a relative error ranging between 4 to 7% (Table 3).



| Date of observation | Number of measurements | Average density ($\rho_{av}$, kg m$^{-3}$) | Standard Deviation ($\sigma_\rho$, kg m$^{-3}$) |
|---|---|---|---|
| Average density on MM snow piles (prior to any action by grooming machines) | | | |
| All sessions | 21 | 437 | 18 |
| Average density on the ski slope (as opened to skiers) | | | |
| 2015-12-04 | 13 | 545 | 31 |
| 2016-01-20 | 8 | 528 | 37 |
| 2016-04-06 | 9 | 618 | 26 |

**Table 3.** Average MM snow density for each session of observations (top) and average snowpack density observed on the ski slope for all three sessions (bottom).

Either on MM snow piles or on the ski slope, the Snow Water Equivalent (SWE, kg m$^{-2}$) was computed for each point of the grid by the relation (2) between snow depth and density:

$$\text{SWE}_{pt} = \text{SD}_{pt} * \rho_{av} \tag{2}$$

The uncertainty on the SWE is computed assuming that the uncertainties on the snow depth ($\sigma_{SD}$) and density ($\sigma_\rho$) are independent and normally distributed (Bevington and Robinson, 2003). The uncertainty $\sigma_{SWE}$ is obtained for each session thanks to the averages SWE$_{av}$ and SD$_{av}$ of the session by the relation (3). The resulting uncertainties $\sigma_{SWE}$ ranged between 20 kg m$^{-2}$ for MM snow observations up to 35 kg m$^{-2}$ on ski slopes.

$$\left(\frac{\sigma_{SWE}}{\text{SWE}_{av}}\right)^2 = \left(\frac{\sigma_{SD}}{\text{SD}_{av}}\right)^2 + \left(\frac{\sigma_\rho}{\rho_{av}}\right)^2 \tag{3}$$

### 2.4 Modelling of snowpack conditions on ski slope

#### 2.4.1 SAFRAN / Crocus-Resort model chain

Crocus Resort is an adapted version of the multilayer physically based snowpack model SURFEX/ISBA-Crocus (Vionnet et al., 2012) and explicitly takes into account the impact of grooming and snowmaking (Spandre et al., 2016). Crocus Resort explicitly solves the equations governing the energy and mass balance of the snowpack on the ski slope. The model time step is 900 s (15 minutes). All simulations in this paper with MM snow production include the impact of grooming on the snow. In French mountain regions, Crocus Resort is usually run using outputs of the meteorological downscaling and surface analysis tool SAFRAN (Durand et al., 1993).

SAFRAN operates on a geographical scale on meteorologically homogeneous mountain ranges (referred to as "massifs") within which meteorological conditions are assumed to depend only on elevation and slope aspect. All simulations in this paper are based on meteorological forcing data from SAFRAN corresponding to Les 2 Alpes site (elevation, slope angle and aspect). We specifically analysed the natural snow conditions provided by SAFRAN-Crocus Resort with in-situ observations on a local





scale from ski patrollers and Automatic Weather Stations (wind, snow/rain elevation limit, precipitation amount). If relevant
we adjusted the SAFRAN meteorological forcing data (amount and snow/rain phase of precipitations) to local conditions for
this site. The deposition rate of dry impurities on the snowpack surface was also adapted to match the natural melting rate at the
end of the season (Brun et al., 1992; Dumont et al., 2012). We also took into account the surrounding slopes of each site and

the consequent shading effects (Morin et al., 2012; Spandre et al., 2016). Last, the wet-bulb temperature was computed from
SAFRAN dry-air temperature and specific humidity using the formulation from Jensen et al. (1990) as described by Spandre
et al. (2016).

### 2.4.2   Performance of the model in simulating the natural snow conditions and wet-bulb temperatures

The computation of the uncertainty on the natural snow water equivalent was based on the simulations results with and without
correction of the forcing data and impurities rate (Section 2.4.1). The RMS of the differences between the simulations and the
in-situ observations are highly reduced (improved simulations) when fitting the meteorological forcing data to the specificities
of the site (Table 4). The final RMS of the differences on SWE (after corrections) is 14 kg m$^{-2}$ and final errors on the
snow depth, SWE and density are similar to Essery et al. (2013), confirming SAFRAN- Crocus provides realistic simulations
of the natural snowpack evolution once adjustments made in albedo (impurities) and forcing data. We therefore assumed
the SAFRAN-Crocus Resort model also provided realistic simulations of the groomed snowpack. We accounted for a larger
uncertainty on the snow water equivalent of the groomed snowpack ($\sigma_{SWE}$ = 30 kg m$^{-2}$ i.e. 0.06 m uncertainty on snow depth
for a 500 kg m$^{-3}$ density, Spandre et al. (2016)).

| Natural snow | RMS Differences | | | Melt-out |
|---|---|---|---|---|
| | SWE (kg m$^{-2}$) | SD (m) | Density (kg m$^{-3}$) | date |
| Observations | N = 6 observations | | | 2016-04-01 |
| SAFRAN - Crocus | 51 | 0.21 | 82 | 2016-04-06 |
| Adjusted SAFRAN - Crocus | 14 | 0.14 | 22 | 2016-04-02 |

**Table 4.** Performance of the snowpack model in simulating the natural snow conditions before and after adjusting the meteorological forcing
data and impurities rate (Section 2.4.1) quantified by the RMS of differences between model and observations.

Besides the natural snow conditions, the cumulated time-span over which wet-bulb temperature fell within specific ranges
was calculated for the MM snow production period i.e. from 20 November 2015 until the 15 March 2016, both from the in-situ
data recorded by the snowgun sensor and the data from SAFRAN (Figure 4). The distribution of the wet-bulb temperature from
SAFRAN meteorological data is very consistent with the T$_w$ distribution from the snowgun sensor.



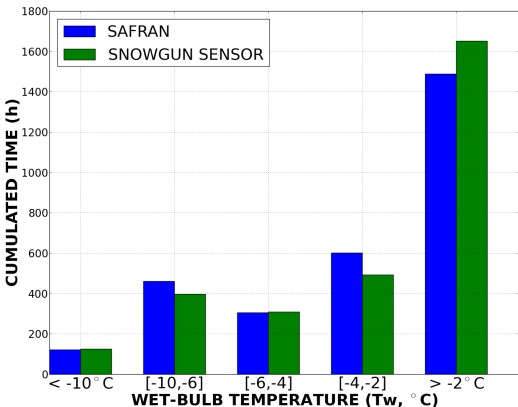

**Figure 4.** Cumulated time-span over which wet-bulb temperature fell within specific ranges, from the in-situ data (snowgun sensor) and SAFRAN (20 November 2015 - 15 March 2016)

### 2.4.3 Snowmaking data

The production period was divided in the model into three distinct periods: before and after the resort opened (5 December 2015) and after the 1 February 2016. The reasons are that the average conditions significantly differ (Table 5) and the ski slope surface opened to skiers was significantly enlarged along the season i.e. modifying the "useful" surface of the ski slope.

5    As a result, we used the water flow recorded by snowmakers and the observed ski slope surface area (Table 5) to force the MM snow precipitation rate in the model which is a constant for each period (expressed in kg m$^{-2}$ s$^{-1}$). The daily time of production was set in the model to match the observed daily production (expressed in kg m$^{-2}$, Figure 2). A wet-bulb temperature of -3.5°C was found as the minimum temperature to trigger snowmaking which afforded to produce the observed amount of MM snow during the first period (21 November - 05 December). Afterwards, the observed MM snow production

10   could be simulated using a trigerring temperature of -5°C.

| Period | Total volume (m$^3$) | Average Water flow (m$^3$ h$^{-1}$) | Average $T_w$ (°C) | Production Surface (m$^2$) | MM Snow Precipitation rate (kg m$^{-2}$ s$^{-1}$) |
|---|---|---|---|---|---|
| 21 Nov. - 05 Dec. | 1629 | 16.2 | - 9.5 | 4063 | 1.11 10$^{-3}$ |
| 05 Dec. - 01 Feb. | 657 | 10.2 | - 6.7 | 6632 | 4.27 10$^{-4}$ |
| 01 Feb. - 01 Apr. | 661 | 11.1 | - 6.9 | 7067 | 4.36 10$^{-4}$ |

**Table 5.** Observed production conditions for the main periods of production before and after the resort opened.



## 3   Results

### 3.1   Observations of MM snow production

Snow piles were usually not much ahead of the snowgun with a significant MM snow depth at the bottom or even at the back
of it (Figure 4a) in consistency with the low wind speed conditions observed in all sessions (Table 6), mainly coming from the
5   East or South-East on average (wind direction not shown). All observed snow piles showed similar geometric patterns (Figures
4a and 4b) resulting in consistent distributions of the snow around the center of the MM snow piles (Figure 4b).

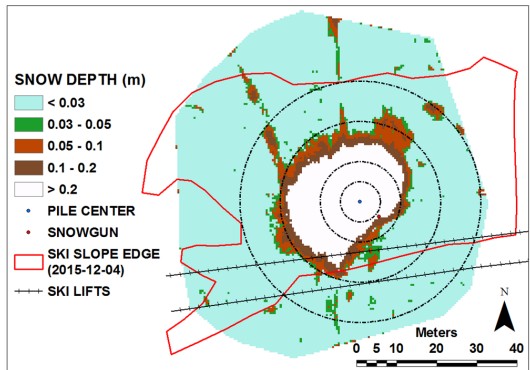

**Figure 4a.** The snow depth raster for the 23 November 2015 production session along with the positions of the snowgun, the center of the
MM snow pile and the concentric circles of radius R = 5, 10, 20 and 30 m. The edge of the ski slope on the 4 November 2015 is also shown.

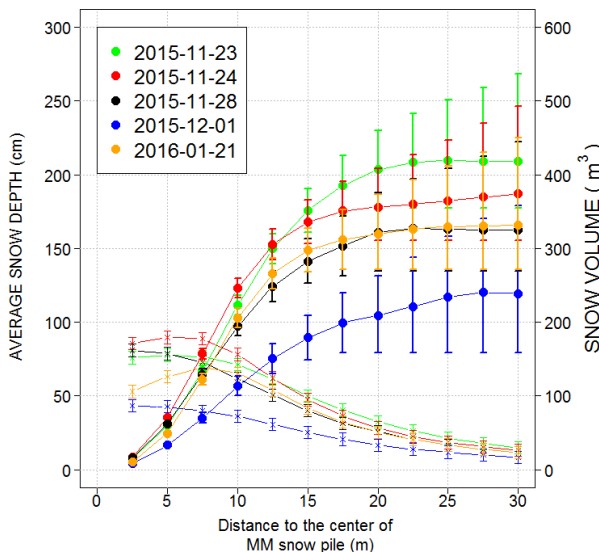

**Figure 4b.** Average snow depth (x) and snow volume (●) within concentric circles around the center of the MM snow pile of radius R from
2.5 to 30 m. The larger the circle the lower the average snow depth and thus the larger the uncertainty on the snow volume.





The average snow depth and the resulting snow volume were calculated for each session of MM snow production within concentric circles around a common fixed point. This point was defined from observations and named "center of MM snow pile" (identical for all sessions, Figure 4a). The equivalent water mass was calculated as the product of the average SWE within the considered circle (relation 2) and the surface of the disk inside the circle, providing kg of water.

| Session | 2015-11-23 | 2015-11-24 | 2015-11-28 | 2015-12-01 | 2016-01-21 |
|---|---|---|---|---|---|
| Water flow ($m^3$ $h^{-1}$) | **18.4** ($\pm$ 1.7) | **18.2** ($\pm$ 1.7) | **17.1** ($\pm$ 1.7) | **13.1** ($\pm$ 1.7) | **12.4** ($\pm$ 1.7) |
| Production duration (h) | 19.6 | 19.2 | 15.8 | 17.3 | 12.0 |
| Wet-bulb Temperature (°C) | **-8.1** ($\pm$ 1.5) | **-8.7** ($\pm$ 1.1) | **-8.5** ($\pm$ 1.4) | **-7.5** ($\pm$ 1.1) | **-7.8** ($\pm$ 1.7) |
| Wind speed (m $s^{-1}$) | **1.82** ($\pm$ 0.8) | **1.06** ($\pm$ 0.48) | **0.53** ($\pm$ 0.53) | **0.56** ($\pm$ 0.47) | **0.53** ($\pm$ 0.58) |
| Quality | "Early bird" | "Early bird" | "Early bird" | "Early bird" | "Care" |
| Recorded water volume ($m^3$) | 361 | 351 | 275 | 227 | 152 |

**Table 6.** Detailed production conditions for every session, with the **average value** ($\pm\sigma$).

## 3.2 Observations of the ski slope snow conditions

The variability of the snow depth (Figure 5) and thus of the associated snow water equivalent (SWE) on the ski slope was significant. The variability (standard deviation) of the SWE values in the study area showed a factor from 3 to 4 with the uncertainty $\sigma_{SWE}$ (Section 2.3, Table 7). Two major observations can be made from the distribution of the snow depth on the ski slope (Figure 5):

- the shape of the MM snow piles was not completely erased by the grooming machines as could be expected. The maximum values of snow depth surrounded the center of the MM snow piles in December and January and was slightly further in April. This may be due to the slow erosion of the snow towards the bottom of the slope by skiers, despite the work made by the grooming machines.

- the initial distribution of the MM snow on the "useful area" defined on the 4 December 2015 could still be noticed on the two latest dates (e.g. the northern and southern edge).

| | Snow Water Equivalent (kg $m^{-2}$) | | |
|---|---|---|---|
| Date of observation | Average $SWE_{av}$ | Spatial Variability (Standard dev.) | Uncertainty $\sigma_{SWE}$ |
| 2015-12-04 | 278 | 87 | 28 |
| 2016-01-20 | 393 | 111 | 35 |
| 2016-04-06 | 501 | 120 | 33 |

**Table 7.** The average SWE observed on the ski slope ($SWE_{av}$) along with the standard deviation of the raster values within the study area (Figure 5) and the uncertainty ($\sigma_{SWE}$) resulting from the computation in Section 2.3.





We performed simulations of the ski slope conditions accounting for the recorded production (100 % water mass, Table 5) or not (0% i.e. groomed only snow) and compared the results with the observations. The average SWE difference between the simulation accounting for MM snow and the observations was 172 kg m$^{-2}$ (RMSD = 204 kg m$^{-2}$) while between the simulation of groomed snow (no production) and the observations the average difference was - 239 kg m$^{-2}$ (RMSD = 282 kg m$^{-2}$).

On the three observations dates, none of the two simulations provided conditions (SD, SWE) within the range of uncertainty of observations. Even though accounting for MM snow production significantly improved the simulation, the differences with observations remained high and suggested the actual amount of MM snow stood between these two simulations.

Based on the observations and the simulations of the natural and groomed snowpacks, we calculated the number of days when the snowpack equivalent water mass exceeded thresholds of 1 kg m$^{-2}$ and 80 kg m$^{-2}$ i.e. respectively the number of days with snow on the ground (Töglhofer et al., 2011) and with suitable conditions for skiing (a minimum of 20 cm of snow with a density of 400 kg m$^{-3}$, Marke et al. (2014)). The following number of days were calculated:

– On the natural snow, the ground was covered by snow for 107 days and the SWE exceeded 80 kg m$^{-2}$ for 48 days of the season,

– On the groomed snowpack (no production), the ground was covered by snow for 133 days and the SWE exceeded 80 kg m$^{-2}$ for 82 days of the season,

– On the ski slope, the ground was covered by snow for 165 days and the SWE exceeded 80 kg m$^{-2}$ for 159 days of the season (estimated from the observed melt-out date and the melting rate between the 6 April and the 3 May 2016).

In Les 2 Alpes ski resort, the ski season extends from the 5 December 2015 until the 30 April 2016 i.e. 148 days during this specific season. The days when the ground was covered by either natural or groomed snow were not consecutive: the snow completely melted in late December and there was no snow during the Christmas holidays in these two cases. Moreover, even though grooming significantly lenghtened the snow cover period, the season length with suitable conditions for skiing was well below the period opened to skiers (82 versus 148 days). The production of MM snow therefore achieved the objectives to provide consecutive days with snow on the ground, to ensure suitable conditions for skiing during Christmas holidays and a sufficient length of the skiing season.



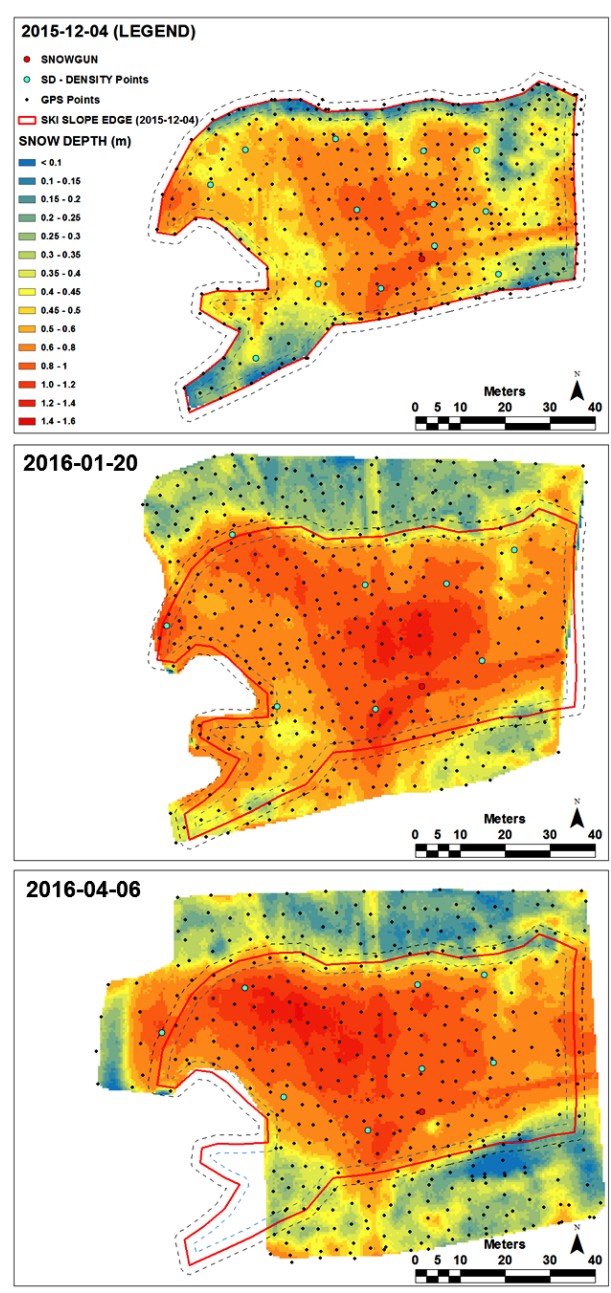

**Figure 5.** Snow depth mapping for the three dates of observations 2015-12-04 (top), 2016-01-20 (center) and 2016-04-06 (bottom).





## 3.3 Definition and computation of the water recovery rate

The Water Recovery Rate (WRR) is defined by the mass balance between the initial mass of water used for production and the resulting mass of MM snow. The WRR ranges between 0 and 1, can be expressed in % and computed either for a MM snow pile prior to any action by grooming machines or for a ski slope snowpack as opened to skiers.

### 5 3.3.1 Water recovery rate from observations of MM snow piles

The MM snow mass (kg) was calculated for single sessions of production from the snow volumes ($m^3$) within concentric circles around the "center" point (Figure 4b) and the MM snow density (kg $m^{-3}$, Table 3). The MM snow mass was further divided by the mass of water used for MM snow production for the given session (Figure 2, Table 6), providing the water recovery rate (WRR, %, Figure 6). Beyond a distance of 20 to 25 m from the center of the snow pile, the MM snow volume does not increase any more while the relative uncertainty further becomes important (over 10%, Figure 6), refraining from taking any conclusion in relation to the recovery rate including those areas. All sessions before the resort opened showed an approximate 20 to 30% water recovery rate within 10 m of distance and 40 to 50% within 20 m (Table 8). The 21 January 2016 session showed a similar behavior with significantly higher WRR (57 and 89% within respectively 10 and 20 m distances). Such differences is further discussed in section 4.

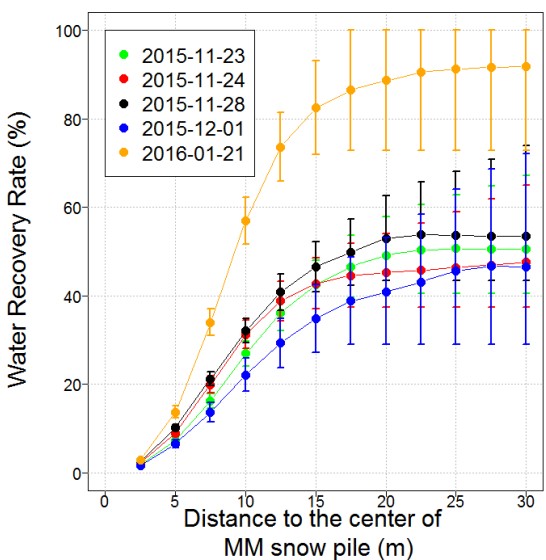

**Figure 6.** The water recovery rate (%) within concentric circles around the center of the MM snow pile. The larger the circle the larger the uncertainty on the snow volume and thus the larger the uncertainty on the water recovery rate.

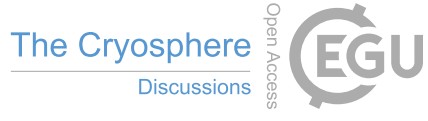

| Session | | 2015-11-23 | 2015-11-24 | 2015-11-28 | 2015-12-01 | 2016-01-21 |
|---|---|---|---|---|---|---|
| Water recovery | **R = 10 m** | **26.9** (± 2.8) | **31.3** (± 3.2) | **32.0** (± 2.7) | **22.1** (± 3.7) | **56.9** (± 5.3) |
| rate (%) | **R = 20 m** | **49.2** (± 8.6) | **45.2** (± 8.8) | **53.0** (± 9.6) | **40.8** (± 12.4) | **88.7** (± 11.4) |

**Table 8.** The water recovery rate within 10 m / 20 m (**Average value** $\pm\sigma$) around the center of the MM snow pile

### 3.3.2 Water recovery rate from observations of the ski slope

The MM snow mass was calculated as the difference between the observed total mass of snow within the edge of the ski slope and the mass of natural snow from the simulated groomed snowpack (Section 2.4.2). The MM snow mass was further divided by the mass of water used for MM snow production up to the date of observations (Figure 2, Table 5), providing the water recovery rate (%).

– On the 4 December 2015, the mass difference within the ski slope edge (4063 m$^2$, Section 2.1) was 974 10$^3$ kg (± 167 10$^3$ kg) i.e. 59.8% (± 10.2%) of the total water mass used for production until this date (1629 10$^3$ kg, Table 5).

– On the 20 January 2016, the mass difference within the ski slope edge (6632 m$^2$, Section 2.1) was 1551 10$^3$ kg (± 306 10$^3$ kg) i.e. 67.9% (± 13.4%) of the total water mass used for production until this date (2286 10$^3$ kg, Table 5).

– On the 6 April 2016, the mass difference within the ski slope edge (7067 m$^2$, Section 2.1) was 1896 10$^3$ kg (± 315 10$^3$ kg) i.e. 64.3% (± 10.7%) of the total water mass used for production until this date (2947 10$^3$ kg, Table 5).

Please note this calculation is based for each date on the total surface of the marked ski slope i.e. a significant part of the early production (before the 5 December 2015) may have fallen beyond the edge of the ski slope as declared opened to skiers on the 4 December 2015 but within the edge of the ski slope as declared opened to skiers on the 20 January 2016 (or even the 6 April 2016). This may partially explain the higher recovery rate on the 20 January and 6 April 2016 compared to the 4 December 2015.

### 3.3.3 Water recovery rate from a crossing of simulations and observations

From the initial simulation with the recorded production (section 3.2) we performed sensitivity tests by running additional simulations using water recovery rates below 1 and computed the RMS of the differences between the simulations and the observations. We decided to use distinct water recovery rates for the first (20 November - 5 December 2015) and the two latest periods, regarding the differences in production conditions (Table 5).

– Considering the first period of production (20 November - 5 December 2015), we simulated snow conditions using water recovery rates from 100% to 30% with a step of 5% and compared the snow conditions (SWE, SD) with the observation on the 4 December 2015. The simulations provided conditions within the range of uncertainty of the observation for water recovery rates of 65, 60 (minimum RMS of differences) and 55% (Table 9).





– From this initial step providing three potential snowpack conditions on the 5 December 2015 (Figure 7), we performed twelve simulations over the second period of the season (after the 5 December 2015) using four distinct water recovery rates of 100, 65, 55 and 45%. Out of these twelve simulations, three provided results within the range of uncertainty for all three dates of observations ($N_U$ = 3) along with the minimum RMS of differences on the SWE (10 to 20 kg m$^{-2}$). Detailed results can be found in Table 9.

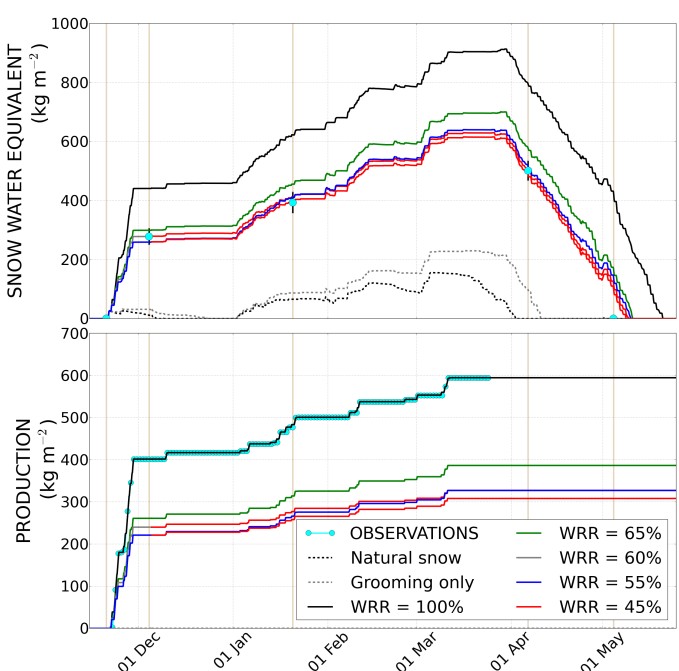

**Figure 7.** Seasonal evolution of the ski slope snowpack. Simulations of the natural snow and groomed natural snow conditions are shown along with the simulations of the ski slope conditions including MM snow production, accounting for water recovery rates (WRR) of 100%, 65% and the three combinations (Table 9) which provided the best agreement with the observations (dots with error bars).

These results suggest that 55 to 65% of the total water mass used for production can be recovered as MM snow within the edge of the ski slope during the first period. This is consistent with the water recovery rates calculated in Section 3.1. The sensitivity test on the water recovery rate did not show any significant difference between the first period of production compared to later in the season. The water recovery rate may even be slightly lower with 45 to 55% of the SWE observed on the ski slope.

The season length was computed from simulations similarly to Section 3.2 for the three combinations of water recovery rates which provided the best agreement with the observations (Table 9). The ground was covered by snow for 170 to 171 days and the SWE exeeded 80 kg m$^{-2}$ for 164 to 166 days of the season, consistently with the observed lengths (Section 3.2). The



bias on the ski season duration and total melt-out date is attributed to a lower melting rate in the snow model compared to the observations: an average - 15.8 to - 16.2 kg m$^{-2}$ day$^{-1}$ for the simulations using the three combinations of water recovery rates (Table 9) with respect to - 17.8 kg m$^{-2}$ day$^{-1}$ for the observations from the 1 April 2016 until the total melt-out date.

| Water Recovery Rate | | | RMS Difference | | | |
|---|---|---|---|---|---|---|
| Period | Periods | $N_U$ | SWE | SD | Density | Melt-out |
| 1 | 2 and 3 | | (kg m$^{-2}$) | (m) | (kg m$^{-3}$) | date |
| OBSERVATIONS | | | N = 3 observations | | | 2016-05-03 |
| 0% | 0% | 0 | 282 | 0.5 | 189 | 2016-04-10 |
| 100% | 100% | 0 | 204 | 0.28 | 50 | 2016-05-15 |
| 65% | 65% | 1 | 51 | 0.03 | 51 | 2016-05-10 |
| 60% | 45% | 3 | 9 | 0.05 | 63 | 2016-05-09 |
| 55% | 55% | 3 | 15 | 0.04 | 54 | 2016-05-09 |
| 55% | 45% | 3 | 11 | 0.07 | 62 | 2016-05-08 |

**Table 9.** Performance of the snowpack model in simulating the ski slope snow conditions. The RMS of the differences between the simulations and the observations are detailed for the 100% and 65% water recovery rates (WRR) simulations, for the three combinations of WRR which provided the best agreement with the observations (Figure 7) and for the simulation of the groomed snowpack (no production, WRR = 0%). Period 1 extends from the 20 November until the 5 December 2015. Period 2 and 3 extend from the 5 December 2015 until the melt-out date.

The interest of both the professional (technical issues, investments) and research (climate change investigations) approaches of the production of snow is to consider the amount of "useful" additional MM snow that can be used on the ski slope. Any difference between the mass of water used for production and the additional snow mass on the ski slope can be considered as water losses in the mass balance. Such losses may be due either to the evaporation and sublimation of the water droplets or snow particles (thermodynamic effects) or to the produced snow falling beyond the edge of the ski slope (mechanical effects). We intend in the following sections to address the impacts of such effects.

## 4 Discussions

### 4.1 Water losses due to thermodynamic effects (evaporation and sublimation)

The losses related to evaporation and sublimation can be calculated for the present study thanks to the linear relationship by Eisel et al. (1988). Although significant changes in snowguns technology may have occured in the last 30 years, this work remains the most detailed on this topic to the best of our knowledge. We may also consider this approach as a "worst case" since the technological evolution presumably evolved for better efficiency. The observed average temperatures of production were respectively - 9.5°C, - 6.7°C and - 6.9°C for the first, second and third periods of production (Table 5), resulting in





respectively 5.84%, 7.9%, 7.7% water losses due to the evaporation of water vapor from droplets and sublimation of ice particles, both during and after their deposition on the ground (Eisel et al., 1988). The overall water loss over the total 2947 m$^3$ used for snowmaking would be 6.7%, i.e. well below the observed differences in the present study. The evaporation and sublimation processes may therefore explain a minor part of the differences reported by either Eisel et al. (1990), Olefs et al.

(2010), Spandre et al. (2016) or observed in the present study. An overall water loss of 40% ($\pm$ 10%) was observed and simulated, among which less than 10% may be due to thermodynamic effects according to Eisel et al. (1988), resulting in additional mechanical water losses of approximately 30% of the total water mass used for MM snow production. The influence of external factors (topography, wind, etc.) proves to be a major concern for water losses and may not be possible to completely assess due to complex dependences.

## 4.2  Water losses due to mechanical effects

Even though the wind conditions were ideal, a significant amount of snow was found at the toe or even at the back of the snowgun (Section 3.1). Since snowguns are usually installed on one side of the ski slope, a part of the production may fall outside the slope, behind the snowgun. The MM snow may also fall beyond the edge of the slope on the opposite side of the snowgun. Hanzer et al. (2014) performed a detailed study of technical snow in an Austrian ski area operating 37 km of ski

slopes for a total surface of 92 ha i.e. an average 25 m wide ski slopes. Spandre et al. (Under review) reported similar data from a survey of French ski resorts with an average ski slopes width of 20 m. The width of a ski slope may have a significant impact on the amount of MM snow falling within the edge of the ski slope regarding the equivalent water masses of MM snow piles within 10 to 20 m from the center point (Table 6). These results also suggest the best position of a snowgun is, if possible, in the middle of the ski slope (as it is the case in some situations).

The surroundings of the ski slope are therefore very important to compute the amount of "useful" MM snow. If the slope can be enlarged (as it is the case for Les 2 Alpes Coolidge slope), the MM snow falling outside the initial edge of the ski slope can be either displaced by the grooming machines or useful for the extension of the slope. In the opposite case where the surroundings have complex topography (e.g. rough surfaces, with rocks) or are covered by vegetation (trees), the amount of snow falling beyond the edge of the ski slope is definitely lost. Consequently the potential of extension of a ski slope is a

significant factor for differences in the MM snow efficiency between slopes (or even resorts). Focusing on this point, the study site may not be representative of the majority of ski slopes. The Coolidge slope is larger (minimum width of 45 m, up to 75 m) than the average dimensions of ski slopes we referred to (Hanzer et al., 2014; Spandre et al., Under review) which makes it a favourable site for the efficiency of MM snow: a maximum amount of the produced snow can be found within the edge of the slope. The total mass of water used for MM snow production also exceeds the usual amounts: Spandre et al. (Under

review) found the usual capacity of water reservoirs was 150 to 190 kg of water per m$^2$ of equipped ski slope with snowmaking facilities with a maximum of 390 kg m$^{-2}$. In the present case, 2947 m$^3$ of water were used for snowmaking (Table 5) over a maximum ski slope surface of 7067 m$^2$ (Section 2.1) i.e. 417 kg m$^{-2}$.

The influence of meteorological conditions on the efficiency of MM snow remains unknown in a large extent and would require further observations to be analysed, referring to findings from Eisel et al. (1988). The meteorological conditions ob-



served in this study appeared as the ideal conditions for the production of MM snow: low wind speed and temperatures (Table 6). Such investigations may be useful for operational purposes to provide objective data on the impact of producing snow in extreme conditions of wind or temperature.

The "Quality" parameter of the MM snow chosen by the professional snowmakers may also have a significant impact on the water recovery rate (Tables 6 and 8, Figure 6). The two sessions on the 21 January 2016 and 1 December 2015 differ mainly due to the parameterization of the "Quality" with significant differences in the WRR. To the best of our knowledge this parameter acts on the volume of the compressed air versus the water volumes within the expelled cloud by the snowgun. There are objective reasons for this parameter to have a significant impact on the water recovery rate. Higher air/water ratio leads to a lower specific humidity of the cloud of droplets and thus a lower gradient with the surrounding ambient air, likely leading to decreasing latent heat exhanges (evaporation and sublimation) and increasing sensible heat transfer i.e. more freezing thanks to a higher surface for heat transfer between the liquid water and the air. Last, the lower water flow means lower speeds of droplets when expelled by the snowgun which therefore have a higher probability to fall within the edge of the ski slope. As an example, on the 28 November 2015 the water mass used for production was $275 \cdot 10^3$ kg, leading to $159 \cdot 10^3$ kg of snow (WRR = 53%, Tables 6 and 8). On the 21 January 2016, the water mass was $152 \cdot 10^3$ kg while the snow mass was $135 \cdot 10^3$ kg (WRR = 89%). Therefore the water mass used on the 28 November 2015 was 1.8 times higher compared with the 21 January 2016 with only 1.08 times more snow mass. Further investigations are strongly needed to improve our understanding of the impact of this parameter and confirm its influence.

## 4.3 Limitations to this work: assessment of water recovery rates and current modelling of ski slope snowpacks

The MM snow mass within the edge of the ski slope was computed from observations (Sections 3.2 and 3.3.2) or simulations (Section 3.3.3) and compared to the recorded mass of water used for production. These computations provided consistent values of the water recovery rate for the first period of production (before the 5 December 2015) with 60% of the total water mass used for production within the edge of the ski slope opened to skiers. Afterwards, the observations of the total mass of snow showed a higher WRR when accounting for the total surface of the ski slope (Section 3.3.2) compared to calculations with the surface limited to the edge of the ski slope on the 4 December 2015 (Section 3.3.3), suggesting a part of the initial production may have fallen beyond the initial edge of the ski slope. This higher WRR could also be due to an improved recovery of individual productions after the 5 December 2015 as suggested by the observations on the MM snow pile on the 21 January 2016. However simulations performed from the initial conditions of the snowpack on the 4 December 2015 suggest that the WRR is lower for the following period than for the first period. Several factors may explain these differences in the WRR either related to objective elements we could not account for or to the weaknesses of the method. We here intend to address such factors:

- The representativity of observations may be questioned. The observations of MM snow piles (Section 3.1) covered 75% of the total mass of water used for production during the first period (1214 out of 1629 m$^3$) while they covered only 11%





of the production after the 5 December 2015 (152 out of 1318 m$^3$). The observation on the 21 January 2016 may not be representative of the whole period of production after the 4 December 2015.

– The difficulty to monitor all human actions on the ski slope (e.g. snow displacement by grooming machines) is a potential source of error. The distribution of snow on the 6 April 2016 (Figure 5) suggests that there was a significant volume of snow displaced from the study area (within the 4 December 2015 edge) to the North-West corner of the ski slope (6 April 2016). Such displacements of snow may explain why the observed snow mass within the initial edge (4 December 2015) did not increase in the second period of production as we expected from the initial snow conditions and the further MM snow productions (after the 4 December 2015).

– Third, the snowpack evolution shows strongly non-linear thermal behaviors (Armstrong and Brun, 2008) which effect might be significant for this study. On one hand the natural and groomed snowpacks in December completely melted, on the other hand the simulations accounting for the production of MM snow did not show a significant loss of equivalent water mass in the same period (Figure 7). As a consequence, the SWE of the groomed snowpack on the 4 December 2015 might not be lost on the ski slope and should be substracted when calculating the mass of MM snow (Section 3.3.2). If accounting for an additional 20 kg m$^{-2}$ equivalent water mass on the 4 December 2015 snowpack, we obtain adjusted water recovery rates of respectively 62.1% and 59.5% for the 20 January and 6 April 2016. These corrected WRR are closer to those computed for the first period (59.8%, Section 3.3.2) and would tend to confirm there is no significant difference in the WRR between the first and the two latest periods of production.

– Last, complementary observations might have reduced the uncertainty over the estimation of the equivalent water recovery rate (an observation was performed on the 2 March 2016 but could not be treated due to a technical failure which made all observations impossible until early April). Since the calculation of the mass of MM snow (Section 3.3.2) depends on the snow water equivalent of the groomed snowpack, observations on ski slopes without production would have been of great help. However all slopes in the surroundings of the study site are equipped with MM snow facilities or under the influence of these facilities. Extra observations on MM snow piles after the 5 December 2015 could have clarified whether the higher WRR observed on the 21 January 2016 were representative of the period or not. Additional observations with different types of snowguns would also have been of interest, although the snowgun used at the observations site is the best-seller of a well-known brand which manages approximately 80% of the snowmaking facilities in French ski resorts (communication from the manufacturer) and may therefore be considered as representative of the current technology.

One dimensional (z-vertical) models feature several limitations for the simulation of ski slopes conditions which are highlighted in the present study through the bias on the total melt-out date related to lower melting rates of the simulations with respect to the observations.

– First, the model can not account for snow / ground partitioning. The variability of the snow depth on the ski slope (Figure 7, Section 3.2) showed there were horizontal heterogeneities of snow properties, either due to the mass transport by skiers





or the partial spreading of MM snow piles by grooming engines. This is particularly obvious when the total melting of the natural (and even groomed) snowpack in December and April made the ski slope an isolated snow patch in a mostly snow-free area with strong edge-effects. In such a situation the energy balance of the snowpack can be significantly affected by the modification of turbulent fluxes (Essery et al., 2006) and horizontal ground fluxes from snow-free areas

to the snow (Lejeune et al., 2007). Since snow free areas have lower albedos than the snow and are not limited to a 0°C maximum temperature, they can become significantly warmer than the surrounding snow and advect heat to the snow through the air (respectively the ground), providing additional sensible heat energy to the snowpack. These two consequences of the snow gound partitioning would enhance the melting rate in the model if they were accounted for, which is not the case.

– Second, the initial rate of impurities in MM snow may also differ from natural snow. The amount of impurities in a snow layer is based in Crocus on both an initial value of impuritites (i.e. initial albedo) and a deposition rate of dry impurities on the snowpack (Brun et al., 1992). There is no reason for the dry deposition to be different between the natural snow and the snow on ski slopes (if at the same place). However the initial amount of impurities in the MM snow could be different than in natural snow: the water used for production is stored in open reservoirs and probably contains more

impurities than the water vapor evolving into atmospheric snow. This could be a reason for a lower albedo of the MM snow which would also enhance the melting rate on ski slopes.

## 5   Conclusions

The present study carried out detailed observations and simulations of the evolution of a ski slope snowpack in Les Deux Alpes ski resort (French Alps), accounting for grooming and the production of MM snow. The snow season was characterized by

a warm, snow deficient early season without natural snow on the ground at the observation site (1680 m.a.s.l) for Christmas holidays. The production of MM snow therefore concentrated in the early season with approximately 50% of the seasonal production realized within one week, late November (Figure 7). The production of MM snow significantly improved the possibility to ski at the observation site with suitable conditions from the opening (5 December 2015) to the closing date of the resort (30 April 2016).

We provided spatial observations of the snow depth and snow water equivalent of MM snow piles (covering 75% of the early season production, five observations in total) and of the ski slope as opened to skiers (three observations in total, Table 1). A high spatial resolution of the snow surface elevation was used (0.25 m² grid) thanks to measurements by a Differential Global Navigation Satellite System (GNSS) and the natural neighbor interpolation method. The related uncertainties were calculated by comparing this method to Terrestrial Laser Scan (TLS) measurements, to Digital Elevation Model (DEM) data

of the bare ground and to probe measurements with a retained uncertainty of 0.042 m on snow depth. The density of snow was measured thanks to snow sampling and weighing, with uncertainties ranging between 4 and 7% (Section 2). The mass of MM snow was deduced from these observations and the difference with the natural snow and was compared to the mass of water used for production, either of a single session (when performing observations on MM snow piles) or up to date



(for ski slope observations). The mass balance between the MM snow mass and the water mass was defined as the water recovery rate. The observations of single sessions of production showed similar distributions around the center of the MM snow pile with approximately 30% WRR within 10 m and 50% within 20 m for production sessions in the early season (Section 3.1). Significantly higher water recovery rates were found for the only session in the heart of the season when a

different parameterization of the snowgun was used by snowmakers. The water recovery rate within the ski slope edge was computed in three occasions with approximately 60% (± 10%) of the water mass used for snowmaking recovered as MM snow (Sections 3.2 and 3.2). The WRR was found relatively constant between observations and simulations and between the different periods of the season. The water losses due to thermodynamic effects were calculated from Eisel et al. (1988) linear approximation with less than 10% of the total water mass either evaporated or sublimated (Section 4). Therefore over 30%

of the water used for snowmaking probably turned to MM snow but could not be recovered within the edge of the ski slope, certainly due to mechanical effects (suspension and erosion by the wind, obstacles, etc.) while the production conditions can be considered as ideal (low wind speed and temperatures, large ski slope).

The water recovery rate of the snowmaking process is therefore a tricky question regarding its likely dependence to both sites characteristics (topography, vegetation) and human decisions (attention to marginal conditions, quality parameter, etc.).

Estimating a single value appears impossible even though the best conditions together (as can be considered the present study) showed a significant fraction of the water used for production was lost for the ski slope. The water recovery rate might have an optimum value corresponding to the best situations. An objective one is definitely the local topography: less than 50% of the water mass can be expected within the edge of a typical ski slope width (approximately 20 to 30 m, Section 4.2) with snowguns on the side and perpendicular to the slope (a typical installation). The authors also hypothetyze that the wind may have a strong

impact on the distances covered by water droplets and ice particles as well as the Quality parameter chosen by professional snowmakers although further investigations of such influences are strongly needed.

Characterizing the actual mass of MM snow that can be recovered on ski slopes from a given mass of water remains a major issue for ski resorts regarding the current development of snowmaking facilities (Spandre et al., 2015) and the related costs of investments and production (Damm et al., 2014). Significant water losses may question the economical interest of

snowmaking for resorts where periods with suitable meteorological conditions are limited in addition to deteriorating factors for the efficiency of MM snow (obstacles e.g. trees, wind).

*Acknowledgements.* The authors wish to thank A. Guerrand (Les Deux Alpes Loisirs) for sharing all details about the management of snow in Les Deux Alpes, the LGGE (Laboratoire de Glaciologie et Géophysique de l'Environnement, Grenoble, France) for provision of the PICO coring auger (D. Six). We also acknowledge the assistance of F. Ousset (Irstea), Y. Deliot and G. Guyomarc'h in the set up and analysis of

field observations, M.Dumont in the parameterization of impurities in Crocus as well as A. Dufour, L. Queno, L. Charrois and J. Revuelto in the fulfillment of observations (all CEN/Météo-France). The Région Rhônes-Alpes is funding Pierre Spandre's PhD. This work has been supported by a grant from "Eau, Neige et Glace" foundation, from the LabEx OSUG@2020 (Investissements d'avenir – ANR10LABX56), and fundings from SO/SOERE GLACIOCLIM, LGGE, IRSTEA, CNRM-GAME/CEN and LTHE.





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
