# Peer review of "Seasonal evolution of a ski slope under natural and artificial snow: detailed observations and modelling"

_The Cryosphere, 2016_

## Referee Comment (RC1) · Anonymous Referee #1 · 13 Oct 2016

General Comments

The authors present a quantitative assessment of the water mass that is lost during production of man-made (MM) snow within a French skiing area using two distinct methods: (i) differential GPS surveys together with snow coring on freshly produced MM snow piles and on a section of a skiing slope and (ii) physically based snow modeling adapted to skiing slope conditions (e.g. accounting for grooming) with and without production of MM snow. The authors conclude that under (the observed) ideal production conditions around 40 % of the water used for MM snow is lost during the production of which 10 % are attributed to thermodynamic effects and 30 % to mechanical and/or technical effects (wind, settings of the snowguns). The quantification of the water mass

lost during MM snow production is not only important for the skiing industry itself (optimizing water and energy costs) but also to raise general awareness for a more responsible use of water and energy resources. For the scientific community this study is highly valuable for two reasons: (i) To my knowledge it is only the second quantitative study after the work of Eisel (1988) which was based on a now outdated technology of snow production, (ii) It allows to correctly quantify the effect of MM snow production in assessments of historical (observed) or future (projected) snow conditions within skiing resorts (related to climate change) by giving the "real" amount of artificial snow present on the slopes instead of an estimated potential production. So far, such loss effects have been very broadly assumed.

In general the manuscript is well structured but partly poorly written and suffers from too much detailed information at some places (please see my detailed comments below) which distracts from the important findings. I congratulate the authors for the serious and detailed treatment of uncertainty concerning their measurement methods and results, which is still too rarely done in geosciences. But I also strongly recommend to use a more rigorous, exact and commonly defined uncertainty terminology (see my detailed comments below). Regarding the role of the thermodynamic effect, the authors should add more details on the method of Eisel (1988) used to estimate sublimation/evaporation of the MM snow and discuss the related uncertainty with regard to the final conclusions of the paper.

Finally, I very much encourage the authors to continue the work in future by investigating in more detail what the influence of some of the questioned points on the total mass loss is (e.g. technical setting of the snow gun ("quality parameter"), detailed thermodynamic effects). Such could e.g. be done by operating a snow gun in a more controlled environment (e.g. parking lot or indoors). This would be very valuable for the scientific community because chances are that you could derive parameterizations that would be transferable in space to be directly used in other related studies.

Therefore I suggest accepting the paper after the points listed in the specific comments

and some minor ones in the technical corrections have been properly addressed by the authors.

Specific Comments (in decreasing order of importance)

(1) Please improve general readability (e.g. English language editor service)

(2) Uncertainty methodology: For clarity and consistency in the scientific community, I very much encourage the Authors to study, use and apply the Guide to the Expression of Uncertainty in Measurement (GUM; JCGM, 2008)) as well as the terminology that is defined therein. In detail it is not clear to the reader what confidence interval the given (combined) uncertainty values you are referring to, what coverage factor you are using etc. (e.g. standard (66% level) vs. expanded (e.g. 95% level) uncertainty).

(3) The authors should provide more details about the Eisel (1988) methodology they use to calculate sublimation/evaporation in Sect. 4.1 and include an estimation of the related uncertainty, also in relation to the main conclusions. Given that this is the main source of potential mass loss beside mechanical effects, this component should be treated with more care in the manuscript!

(4) To my mind, the manuscript would very much profit from being more concise at some places:

a. Sect. 2.2.3 (Evaluation of related uncertainties on snow depth) should be strongly shortened. I would suggest to move everything from p6 to p8 (including) to an appendix or additional material (also Fig. 2a,b and 3 and Table 2). The essential information is written on p9.

b. Most of the information of Sect.3.3.2 (Water recovery rate from observations of the ski slope) could be put in a table, this would be much clearer.

c. The abstract and the conclusions could be shortened with the same essential information content.

(5) It would be interesting to have some photos of the produced and measured snow piles and the snow guns and where the produced snow is found (toe/back of the snowgun,...) as e.g. additional material.

(6) More technical details about the snow guns used for MM snow production would be helpful as well (e.g. air-water gun or fan gun) and an estimation whether your results are applicable for other types of snow guns. In order of appearance

(7) P 4 L 21-22: "We expected...." I do not understand this sentence.

(8) P10: please add more details about the weighting method to measure snow density on MM snow piles and the related uncertainty.

(9) P11 L4: it is not clear to me how the uncertainty on the density is derived.

(10) P12 L20: please describe the snow gun sensor (type of sensor, accuracy,...)

(11) P15: The "quality" parameter should be clearer introduced and explained in the methods section.

(12) P21 L8: what about the wind erosion of already deposited MM snow? could this be an additional source of mass loss ? If not, please explain here why not. And what about melting? If you can exclude it add it here for completeness as well and argue why.

Technical corrections (general)

Whenever referring to a date in your manuscript it should read "on 4 December" and NOT "on the 4 December"... I would prefer "equation" instead of "relation".

Technical corrections (in order of appearance)

P1 L20: is snow a "material"?

P1 L20-21: "...encouraging ski lifts operators to an increasing amount of technical methods of snow.. "

P2 L1: "ski resort stakeholders"

P2 L6: "…Machine Made (MM) snow mass on ski slopes…"

P2 L11: "They found an average of 6% of water loss…"

P2 L15: "…a maximum of 4 m$^3$…"

P2 L16: "…and with a MM snow technology". Abbreviation MM was already introduced.

P2 L25: "…the efficiency of MM snow production…"

P2 L29-30: Please clarify what you mean by "dedicated sessions" as this terminus is used several times afterwards.

P4 L6: "(wet-bulb temperature of the air and wind conditions)"

P6 L23: "…interpolated GPS points with the TLS points…"

P6 L31: There is something wrong with "…determined help to a total station"

P7 L2: "… An average elevation difference of -0.0012 m was measured…"

P13 L8: I would prefer to call it a "threshold wet-bulb temperature for snowmaking"

P16 L12-17: I would replace "on the natural/groomed snowpack" with e.g. "concerning the natural/groomed snowpack…"

P22 L8-9: "…not be possible to be completely assessed due to complex…"

Figures and Tables

Figure 2. The light brown vertical lines are hardly visible

Figure 4a. Think of adding more color to the colormap, values > 0.2 m are not visible, the position of the snow gun is not visible.

Figure 4b. Please use SI units (left ordinate), the average snow depths (x) are hardly

visible.

Figure 5. The position of the snowgun is hardly visible. Caption: add the information what area this data shows (skiing slope!?).

References

Eisel, L. M., Mills, K. D., and Leaf, C. F.: Estimated consumptive loss from man made snow, JAWRA Journal of the American Water Resources Association, 24, 815–820, doi:10.1111/j.1752-1688.1988.tb00932.x, 1988

Joint Committee for Guides in Measurements (JCGM): Evaluation of measurement data – Guide to expression of uncertainty in measurement, JCGM 100:2008, GUM 1995 with minor corrections, available at: http://www.bipm.org/utils/common/documents/jcgm/JCGM_100_2008_E.pdf, 2008.

---

## Referee Comment (RC2) · F. Wolfsperger (Referee) · 2 Nov 2016

general comments

Spandre et al. present unique data on the mass balance of snowmaking which firstly quantifies the net water losses (10 to 50%) with high validity. The findings are highly relevant for ski resorts and the snow making industry as well as for the scientific community as several publications describe snow making models without taking account mass losses.

The design of the study and the methods are adequate: An area of a ski piste is monitored from the first snowfalls to the complete melt of the snowpack by measuring

the snow surface (ground was measured too) and snow density at several points in time to quantify the snow mass on the ground of single snow making events as well as the total snow accumulation and its redistribution throughout the season. The water flow of the snow making system was recorded continuously providing the used water mass and led to the snow-water-mass ratio.

The GNSS method was properly validated comparing to TLS method. The propagation of uncertainties into the snow volume calculation was considered but some statistical methods were unsuited (see below). SAFRAN-crocus model chain was run with precipitation correction, a snowmaking and grooming module. In the methods section, the authors miss to clearly outline the use of model in their study. The explanation comes later in the result section: First, the model was used to answer the question how good (how much snow over time of season) the piste would have been - a) without snowmaking and b) without snowmaking and without grooming compared to c) the observation. Second, the model was used to determine the mass losses of snow making throughout the whole season by comparing the measured total amount of snow with the simulated amount of snow excluding the produced MM snow in the model. Third, mass losses are implemented into the snow making module of the model to best meet the measured seasonal snow mass evolution of the piste evolution.

The weakness of the manuscript becomes obvious: Two different objectives are mixed up leading to a confusing structure. a) To determine snowmaking mass losses, two approaches are presented: one by measurements of single snowmaking events, the other analyzes the whole season and uses modeling and measurements. b) Conclusions on the seasonal evolution of the ski slope are more a side product of a).

In the introduction, the relevance of the study was also argued by the snow making efficiency and not by the seasonal evolution of the piste which is pretty known, even among practitioners as DGPS-snow heights measurements are state of the art (in a few resorts). If focus should be on the seasonal evolution of the snow pack, additional parameters should be presented like the mentioned albedo influencing factors dirt or

SSA, snow wetness etc.

So, to overcome those problems I would recommend minor revisions: - changing the title that it fits to the relevant finding of the study - restructure methods and results *WRR(mass loss) by observations of single snowmaking events *WRR by observation & modelling of seasonal snow accumulation *indirect WRR estimation: best fit of modelled snow accumulation to observations

specific comments

p1, l.13: ...we also addressed thermodynamic effects... Eisel did this with an over-simplified formulation p4: an overview on the test site would be nice (from above) p7, l.5: Shapiro Wilk Test is suitable for n<=2000 (Royston, 1982). I guess with n>8000 or >16000 the test would never accept H0 (accepting it's not different to normality) p.9, l.9-11: and (1): which uncertainties are combined how? (sorry, didn't get it...) p.10, l.6: ...relative error of 4%... wouldn't call this error. It's rather natural variability of MM snow. Better: ... showed a weak variation of 4% (sigma) p.10, l.2-9: generally snow density is quite a sensitive term when determining snow mass as MM snow density can vary from 350 to 600 kg/m3 mainly driven by liquid water content depending on meteo and snow producer props (flow...) and water temp. LWC can also vary along the snow pile. Lower snow quality adjustment at the snow maker simply increases the water flow and usually leads to higher LWC and so density. Taking an average density about all experiments has to be justified (well, you did with sd=4%)but it would also be nice to explain why density could vary. May also give some information on where at the pile density was measured (e.g max height). I see this point p.12, l.21: ...is very consistent...maybe show difference of total snow making hours - looks like a few days and more than 10% p. 13, Fig.4: classes would be better started with >-3.5°C your start threshold p.13: how many snowmakers were used and which type? p.14: Fig 4a) at N side of the pile two "lines"(snow accumulations) are visible. Is there an explanation. How the center of pile was defined (max height?) p.14: Fig 4b)Not sure if geometric patterns can be recognize by the visualization. Error bars should be described (A(R)*SD ?) p.15: Tab6: why is sigma of water flow constant? why there is no sigma for total water vol. p.16: why is the groomed piste so much better - it has the same snow mass, right? just a smaller volume by compacting? SSA and so albedo should be higher without grooming? wind drift? thermal cond? p.18: maybe define a WRR(R_max) with R_max = dv/dR =0 p.19: Tab.8 sigma is not explained p.19: better explain how WRR and its uncertainty is calculated with Formula where simulated and measured mass terms should be indexed p.19: l.6-12: I would prefer a table for these results p.19: l.18: see general comments: what is the aim? methods explained with results. p.21: Tab.9: what is Nu? RMSE is not showing if snow mass is smaller or larger p.21: 12ff: explanations on the physics of snow making are missing and why the eisel approach is trusted or maybe not. p.22: would prefer wind drift for mechanical effects. p.23: speed of droplets depends on pressure not on flow. The effect might be due to smaller droplets (less mass and impulse) connected to different nozzles. Quality acts on water flow sometimes droplet size, so better freezing due to less mass concentration in the snow cloud. p.25, l.4: see Mott et al., 2011 p.25ff, l.27: 0.5m grid, most of the conclusion is a summary

---

## Author Comment (AC1) · 10 Nov 2016

Dear editor, dear reviewers,

i and my colleagues and co authors acknowledge your consideration and useful comments and suggestions on our paper "Seasonal evolution of a ski slope...". We will shortly reply to your detailed reviews and submit a revised manuscript. Regarding i will defend my PhD thesis the 5 December, we woud be very grateful if the submission date was extended to the 20 December 2016.

Yours sincerely, Pierre Spandre on behalf of all co-authors

---

## Referee Comment (RC4) · Anonymous Referee #3 · 16 Nov 2016

General comments:

The paper presents a quantitative assessment of the water mass that is lost during production of man-made (MM) snow within a French skiing area, using two distinct methods. The authors conclude that under the recorded production conditions around 40 % of the water used for MM snow is lost during the production, among which 10 % are attributed to thermodynamic effects and 30 % to mechanical and/or technical effects (wind, topography....). I suggest accepting the paper but some moderate/major revisions are needed according to the listed specific comments.

Specific comments:

[Figure]

In general the manuscript is well structured but requires a general overview by a an english mother tongue. Moreover it should focus mainly on the main outcomes of the research, with a reduction of some sections that now are too wordy. In particular the abstract and the conclusions should be shortened.

Specific comments are listed below:

Pag 1 line 25: I think that snowmaking is not only related to the mitigation/adaptation of the effects of climate change but also to face the natural variability of snow conditions. Keep in mind the concept of natural variability as you reported in the abstract (Pag 1 line2).

Pag 2 line 5: See also: Signatures of Evaporation of Artificial Snow in the Alpine Lower Troposphere (SEASALT) Geophysical Research Abstracts, Vol. 10, EGU2008-A-11002, 2008 SRef-ID: 1607-7962/gra/EGU2008-A-11002 EGU General Assembly 2008

Pag 2 line 21: How did they explain such differences between fan and air water guns?

Pag 2 line 23: How the vegetation may influence the efficiency of MM snow?

Pag 2 line 24: Here the authors report as external factors wind, topography and vegetation. At pag 1 line 18 you reported human decisions and not vegetation. Please here and in the all text consider the same external factors.

Pag 2 line 30: Among the available data on the snow production why you don't consider the relative humidity in air?

Pag 3 line 4: What do you mean with the term important? Do you refer to the number of skiers that ski along the slope?

Pag 9 line 10: At what depth did you measure the snow density? Do you assume that the snow density was constant at the different depths in the snow piles? I think that the snow density could be higher at greater depths in the snow piles. What are the main

results from the application of the PICO coring auger (Pag 9 line 13).

Pag 9 line 11: Why you could not perform the snow density measurement?

Pag 11 line 3: Do you assume a higher melting rate due to the reduction of albedo caused by the deposition of dry particles?

Pag 20 line 10: I think that the authors should provide more details about the methodology elaborated by Eisel (1988) to calculate sublimation/evaporation. Since this is the main source of potential mass loss beside mechanical effects, this subject shoud be better clarified in the whole paper.

Pag 21 line 31: Among the ideal conditions I would add also the low water content in air.

Pag 23 line 6: Do you mean that in December you had an extreme meteorological event? Please take into account that December 2015 was exceptionally mild with little snow. How this extreme meteorological event affect the snowmaking in the study area? You reported this point in the conclusions but not in the Discussion section.

Technical corrections: Pag 3 line 4: add . after a.s.l.
* * *

---

## Author Response (AR1)

**General Comments to Editor**

We thank Fabian Wolfsperger and the two additional reviewers for their thorough evaluation of our work. We did our best to improve our manuscript based on their comments, including a deep review of the language by means of a review by a native professional proofreader. The structure of the manuscript was also modified as suggested by the reviewers and some sections shortened. Major modifications are highlighted in blue.

We hope this new version of the manuscript is now suitable and fits the usual expectations for publication in The Cryosphere.

Best Regards,
Pierre Spandre, on behalf of all co-authors

**Reviewer #1 Anonymous**

**General Comments**

The authors present a quantitative assessment of the water mass that is lost during production of man-made (MM) snow within a French skiing area using two distinct methods:

(i) differential GPS surveys together with snow coring on freshly produced MM snow piles and on a section of a skiing slope and

(ii) physically based snow modeling adapted to skiing slope conditions (e.g. accounting for grooming) with and without production of MM snow. The authors conclude that under (the observed) ideal production conditions around 40 % of the water used for MM snow is lost during the production of which 10 % are attributed to thermodynamic effects and 30 % to mechanical and/or technical effects (wind, settings of the snowguns). The quantification of the water mass lost during MM snow production is not only important for the skiing industry itself (optimizing water and energy costs) but also to raise general awareness for a more responsible use of water and energy resources.

For the scientific community this study is highly valuable for two reasons: (i) To my knowledge it is only the second quantitative study after the work of Eisel (1988) which was based on a now outdated technology of snow production, (ii) It allows to correctly quantify the effect of MM snow production in assessments of historical (observed) or future (projected) snow conditions within skiing resorts (related to climate change) by giving the "real" amount of artificial snow present on the slopes instead of an estimated potential production. So far, such loss effects have been very broadly assumed.

In general the manuscript is well structured but partly poorly written and suffers from too much detailed information at some places (please see my detailed comments below) which distracts from the important findings. I congratulate the authors for the serious and detailed treatment of uncertainty concerning their measurement methods and results, which is still too rarely done in geosciences. But I also strongly recommend to use a more rigorous, exact and commonly defined uncertainty terminology (see my detailed comments below). Regarding the role of the thermodynamic effect, the authors should add more details on the method of Eisel (1988) used to estimate sublimation/evaporation of the MM snow and discuss the related uncertainty with regard to the final conclusions of the paper.

Finally, I very much encourage the authors to continue the work in future by investigating in more detail what the influence of some of the questioned points on the total mass loss is (e.g. technical setting of the snow gun ("quality parameter"), detailed thermodynamic effects). Such could e.g. be done by operating a snow gun in a more controlled environment (e.g. parking lot or indoors). This would be very valuable for the scientific community because chances are that you could derive parameterizations that would be transferable in space to be directly used in other related studies. Therefore I suggest accepting the paper after the points listed in the specific comments and some minor ones in the technical corrections have been properly addressed by the authors.

**We thank Reviewer #1 for his/her thorough evaluation of our work.**

**Specific Comments (in decreasing order of importance)**

**Done** (1) Please improve general readability (e.g. English language editor service)
**[1A]** The manuscript was corrected by a native, professional proofreader

**Done** (2) Uncertainty methodology: For clarity and consistency in the scientific community, I very much encourage the Authors to study, use and apply the Guide to the Expression of Uncertainty in Measurement (GUM; JCGM, 2008)) as well as the terminology that is defined therein. In detail it is not clear to the reader what confidence interval the given (combined) uncertainty values you are referring to, what coverage factor you are using etc. (e.g. standard (66% level) vs. expanded (e.g. 95% level) uncertainty).
**[1B]** We thank Reviewer #1 for this comment.
We defined the error on snow depth as the standard deviation of differences within pixels (0.5x0.5) by the laser scan method (2.2.3). The uncertainty on the density was considered as the standard deviation of measurements (Table 3). Therefore the combined errors are based on the "standard" 68% confidence interval ($\pm\sigma$).
We outlined this point in section 2.3.

**Done** (3) The authors should provide more details about the Eisel (1988) methodology they use to calculate sublimation/evaporation in Sect. 4.1 and include an estimation of the related uncertainty, also in relation to the main conclusions. Given that this is the main source of potential mass loss beside mechanical effects, this component should be treated with more care in the manuscript!
**[1C]** Eisel et al. (1988) reported experiments with an average value of water losses of 5.8% and a 95% confidence interval from 2.5 to 9.1%, i.e. an uncertainty estimated below 3%.
Based on our data, we computed an average 6.7% water losses due to thermodynamic effects thanks to the equation by Eisel et al. (1988) in section 4.1 and further concluded that "less than 10% may be due to thermodynamic effects according to Eisel (1988)" therefore accounting for the uncertainty computed by Eisel (1988).

We detailed the uncertainty defined by Eisel et al. (1988) i.e. 3% in section 4.1 but did not provide further details on the methodology since the paper by Eisel et al. (1988) is available online for reading and to keep the manuscript as concise as possible.

(4) To my mind, the manuscript would very much profit from being more concise at some places:
**Done.** a. Sect. 2.2.3 (Evaluation of related uncertainties on snow depth) should be strongly shortened. I would suggest to move everything from p6 to p8 (including) to an appendix or additional material (also Fig. 2a,b and 3 and Table 2). The essential information is written on p9.
**[1D]** Section 2.2.3 shortened. Now Fig 2a,b, 3 in Appendix B.

**Done** b. Most of the information of Sect.3.3.2 (Water recovery rate from observations of the ski slope) could be put in a table, this would be much clearer.
**[1E]** We created the Table 9

c. The abstract and the conclusions could be shortened with the same essential information content.
**[1F]** Abstract is now 14 lines long (versus 18 initially)
Introduction is now 31 lines long (versus 40 initially)
Conclusion is now 33 lines long (versus 42 initially)

Overall, the paper is 23.5 pages long versus 26 initially. An appendix was created to remove details from the methods section.

**Done** (5) It would be interesting to have some photos of the produced and measured snow piles and the snow guns and where the produced snow is found (toe/back of the snowgun) as e.g. additional material.
**[1G]** Added appendix A with
- the situation of the observations site from above.
- A picture in the morning of the 27 november with MM snow piles.

**Done** (6) More technical details about the snow guns used for MM snow production would be helpful as well (e.g. air-water gun or fan gun) and an estimation whether your results are applicable for other types of snow guns. In order of appearance
**[1H]** Indeed, we omitted to indicate we used a single air/water gun for all observations. This is now detailed in section 2.1.

Because we do not have any partnership or any relationship with the manufacturer, we decided not to indicate the model and brand of the snowgun. However this air/water gun is the most sold model in France and we expect that many of our results can be applicable to other types of snowguns (e.g. mechanical effects, see section 4.3).

**Done** (7) P 4 L 21-22: "We expected." I do not understand this sentence.
Reformulated

**Done** (8) P10: please add more details about the weighting method to measure snow density on MM snow piles and the related uncertainty.
(9) P11 L4: it is not clear to me how the uncertainty on the density is derived.
See section 2.3

**Done** (10) P12 L20: please describe the snow gun sensor (type of sensor, accuracy)
**[1I]** See comment **[1H]**
Based on communications with the snowgun manufacturer, the uncertainty on water volumes used for snowmaking was neglected (below 1% according to the manufacturer). Now detailed in section 2.1.

(11) P15: The "quality" parameter should be clearer introduced and explained in the methods section.

**Done** (12) P21 L8: what about the wind erosion of already deposited MM snow? could this be an additional source of mass loss ? If not, please explain here why not. And what about melting? If you can exclude it add it here for completeness as well and argue why.
It is very unlikely that MM snow may be drifted by the wind due to both the density (over 400 kg/m3) and the cohesion of snow grains (capillarity/refrozen bridges).
Added an explanation in section 4.2.

**Technical corrections (general)**
**Done** Whenever referring to a date in your manuscript it should read "on 4 December" and
NOT "on the 4 December"

**Done** would prefer "equation" instead of "relation".

**Technical corrections** (in order of appearance)

**Done** P1 L20: is snow a "material"?
Removed

**Done** P1 L20-21: "...encouraging ski lifts operators to an increasing amount of technical methods of snow.. "

**Done** P2 L1: "ski resort stakeholders"

**Done** P2 L6: "…Machine Made (MM) snow mass on ski slopes…"

**Done** P2 L11: "They found an average of 6% of water loss…"

**Done** P2 L15: "…a maximum of 4 m3…"

**Done** P2 L16: "…and with a MM snow technology". Abbreviation MM was already introduced.

**Done** P2 L25: "…the efficiency of MM snow production…"

**Done** P2 L29-30: Please clarify what you mean by "dedicated sessions" as this terminus is used several times afterwards.

**Done** P4 L6: "(wet-bulb temperature of the air and wind conditions)"

**Done** P6 L23: "…interpolated GPS points with the TLS points…"

**Done** P6 L31: There is something wrong with "…determined help to a total station"

**Done** P7 L2: "…An average elevation difference of -0.0012 m was measured…"

**Done** P13 L8: I would prefer to call it a "threshold wet-bulb temperature for snowmaking"

**Done** P16 L12-17: I would replace "on the natural/groomed snowpack" with e.g. "concerning the natural/groomed snowpack…"

**Done** P22 L8-9: "…not be possible to be completely assessed due to complex…"

Figures and Tables

**Done** Figure 2. The light brown vertical lines are hardly visible

**Done** Figure 4a. Think of adding more color to the colormap, values > 0.2 m are not visible, the position of the snow gun is not visible.

**Done** Figure 4b. Please use SI units (left ordinate), the average snow depths (x) are hardly visible.

**Done** Figure 5. The position of the snowgun is hardly visible. Caption: add the information what area this data shows (skiing slope!?).
Now Appendix A shows the observations site from above. Detailed in caption.

**References**

Eisel, L. M., Mills, K. D., and Leaf, C. F.: Estimated consumptive loss from man made snow, JAWRA Journal of the American Water Resources Association, 24, 815–820, doi:10.1111/j.1752-1688.1988.tb00932.x, 1988

Joint Committee for Guides in Measurements (JCGM): Evaluation of measurement data – Guide to expression of uncertainty in measurement, JCGM 100:2008, GUM 1995 with minor corrections, available at: http://www.bipm.org/utils/common/documents/jcgm/JCGM_100_2008_E.pdf, 2008.

**Reviewer #2 Fabian Wolsfperger**

**general comments**

Spandre et al. present unique data on the mass balance of snowmaking which firstly quantifies the net water losses (10 to 50%) with high validity. The findings are highly relevant for ski resorts and the snow making industry as well as for the scientific community as several publications describe snow making models without taking account mass losses.

The design of the study and the methods are adequate: An area of a ski piste is monitored from the first snowfalls to the complete melt of the snowpack by measuring the snow surface (ground was

measured too) and snow density at several points in time to quantify the snow mass on the ground of single snow making events as well as the total snow accumulation and its redistribution throughout the season. The water flow of the snow making system was recorded continuously providing the used water mass and led to the snow-water-mass ratio.

The GNSS method was properly validated comparing to TLS method. The propagation of uncertainties into the snow volume calculation was considered but some statistical methods were unsuited (see below). SAFRAN-crocus model chain was run with precipitation correction, a snowmaking and grooming module. In the methods section, the authors miss to clearly outline the use of model in their study. The explanation comes later in the result section: First, the model was used to answer the question how good (how much snow over time of season) the piste would have been - a) without snowmaking and b) without snowmaking and without grooming compared to c) the observation. Second, the model was used to determine the mass losses of snow making throughout the whole season by comparing the measured total amount of snow with the simulated amount of snow excluding the produced MM snow in the model. Third, mass losses are implemented into the snow making module of the model to best meet the measured seasonal snow mass evolution of the piste evolution.

The weakness of the manuscript becomes obvious: Two different objectives are mixed up leading to a confusing structure. a) To determine snowmaking mass losses, two approaches are presented: one by measurements of single snowmaking events, the other analyzes the whole season and uses modeling and measurements. b) Conclusions on the seasonal evolution of the ski slope are more a side product of a).

In the introduction, the relevance of the study was also argued by the snow making efficiency and not by the seasonal evolution of the piste which is pretty known, even among practitioners as DGPS-snow heights measurements are state of the art (in a few resorts). If focus should be on the seasonal evolution of the snow pack, additional parameters should be presented like the mentioned albedo influencing factors dirt or SSA, snow wetness etc.

**We thank Reviewer #2 Fabian Wolsperger for his thorough evaluation of our work. Some of the replies to his questions and issues are common to the replies to Reviewer #1 and were not repeated.**

So, to overcome those problems I would recommend minor revisions:
**Done**  - changing the title that it fits to the relevant finding of the study

**Done**  - restructure methods and results
        *WRR(mass loss) by observations of single snowmaking events
        *WRR by observation & modelling of seasonal snow accumulation
        *indirect WRR estimation: best fit of modelled snow accumulation to observations
        **[2A]** We retitled the sections 3.1 and 3.2 to make the difference between "single events" and "seasonal snow accumulation" measurements. Water recovery rate (WRR) is defined in section 2.5 and computed in section 3.3 where we also retitled subsections to make it clearer on which measurements are used to estimate the WRR.
        Now created section 2.5 "Definition and computation of the WRR" to remove all methods initially in section 3.3.

**Specific comments**
**Done**  p1, l.13: …we also addressed thermodynamic effects… Eisel did this with an oversimplified formulation
        **[2B]** See comment [1C].

To the best of our knowledge, the paper by Eisel (1988) is the most complete on this question so far, combining experiments (mass balance) and computation (energy balance).

**Done**  p4: an overview on the test site would be nice (from above)
**[2C]** See comment **[1G]**, we added appendix A.

**Done**  p7, l.5: Shapiro Wilk Test is suitable for n<=2000 (Royston, 1982). I guess with n>8000 or >16000 the test would never accept H0 (accepting it's not different to normality)
**[2D]** indeed,

Shapiro Wilk test is suitable for a sample N < 5000 (Royston (1982) corresponds to the initial development limited to N=2000 and was further extended by Royston (1995) to N=5000, see also http://stat.ethz.ch/R-manual/R-devel/library/stats/html/shapiro.test.html
We therefore used this test on 1000 sub samples of N=5000 points among the 8072 (respectively 16179 points) and:
1/ checked all tests provided the same conclusions (p-value<0.05 and w>0.9)
2/ indicated the average p-value and w in the Table 5.2
We outlined this method in the Appendix B.

**Done**  p.9, l.9-11: and (1): which uncertainties are combined how? (sorry, didn't get it…)
**[2E]** The snow depth is defined as the difference in elevations of snow surfaces before/after the snow production. Therefore the uncertainty on snow depth is deduced from the uncertainty on the elevations of snow surfaces.
We outlined this point in the text.

**Done**  p.10, l.6: …relative error of 4%... wouldn't call this error. It's rather natural variability of MM snow. Better: … showed a weak variation of 4% (sigma)
Reformulated

**Done**  p.10, l.2-9: generally snow density is quite a sensitive term when determining snow mass as MM snow density can vary from 350 to 600 kg/m3 mainly driven by liquid water content depending on meteo and snow producer props (flow…) and water temp. LWC can also vary along the snow pile. Lower snow quality adjustment at the snow maker simply increases the water flow and usually leads to higher LWC and so density. Taking an average density about all experiments has to be justified (well, you did with sd=4%) but it would also be nice to explain why density could vary. May also give some information on where at the pile density was measured (e.g max height). I see this point
**[2F]** The MM snow density was measured in several points of the snow pile. This is now detailed in the text.

p.12, l.21: …is very consistent…maybe show difference of total snow making hours - looks like a few days and more than 10%
p. 13, Fig.4: classes would be better started with >-3.5°C your start threshold
Figure 3 shows the cumulated time span with Tw temperature between specific thresholds. These are not snowmaking hours.

**Done**  p.13: how many snowmakers were used and which type?
See comment **[1H]**

**Done** p.14: Fig 4a) at N side of the pile two "lines"(snow accumulations) are visible. Is there an explanation. How the center of pile was defined (max height?)

**[2G]** The center of the MM snow pile was defined on the field for every session of observations as the maximum height. The location showed a very weak variation, we therefore considered the same point for the present analysis.

**Done** p.14: Fig 4b)Not sure if geometric patterns can be recognize by the visualization. Error bars should be described (A(R)*SD ?)

**[2H]** Now described in section 3.1 (figure 4b) and 3.3.1 (figure 6)

p.15: Tab6: why is sigma of water flow constant? why there is no sigma for total water vol.

See comment **[1I]**

p.16: why is the groomed piste so much better - it has the same snow mass, right? just a smaller volume by compacting? SSA and so albedo should be higher without grooming? wind drift? thermal cond?

**[2I]** If this comment refers to the season length (l18 - 24), the physical explanation appeared to us as beyond the scope of this paper and is related to the increased density of the groomed snowpack, leading to an increased thermal conductivity. This is the main driver for the differences in the seasonal evolution of natural vs. groomed snowpacks. An additional publication on this question can be found below.

Spandre, P., S. Morin, M. Lafaysse, Y. Lejeune, H. François and E. George-Marcelpoil, Integration of snow management processes into a detailed snowpack model, Cold Reg. Sci. Technol., 125, 48-64,doi:10.1016/j.coldregions.2016.01.002, 2016.

**Done** p.19: Tab.8 sigma is not explained

**Done** p.19: better explain how WRR and its uncertainty is calculated with Formula where simulated and measured mass terms should be indexed

**[2J]** Now WRR defined in methods (section 2.5)

Equations 4 and 5

**Done** p.19: l.6-12: I would prefer a table for these results

See comment **[1E]**

**Done** p.19: l.18: see general comments: what is the aim? methods explained with results.

See comment [2J], the WRR is defined in section 2.5.

We also detail that the determination of the WRR will be based on both observations and simulations. All methods initially in section 3.3.3 now in 2.5.2.

**Done** p.21: Tab.9: what is Nu? RMSE is not showing if snow mass is smaller or larger

We added and additional definition in the caption of Table 10.

p.21: 12ff: explanations on the physics of snow making are missing and why the eisel approach is trusted or maybe not.

See comment **[1C]**

p.22: would prefer wind drift for mechanical effects.

Mechanical effects are not limited to wind drift. We therefore did not change the title for subsection 4.2.

p.23: speed of droplets depends on pressure not on flow. The effect might be due to smaller droplets (less mass and impulse) connected to different nozzles. Quality acts on water flow sometimes droplet size, so better freezing due to less mass concentration in the snow cloud.

The water flow of a snowgun depends on both the number of used nozzles and the water pressure in the snowgun.

For a given number of nozzles which expel water, a higher water flow is provided by a higher input water pressure and therefore results in a higher speed of droplets.

p.25, l.4: see Mott et al., 2011

We could not find this reference without additional details

**Done** p.25ff, l.27: 0.5m grid, most of the conclusion is a summary
* * *
**Reviewer #3 Anonymous**

**General comments:**

The paper presents a quantitative assessment of the water mass that is lost during production of man-made (MM) snow within a French skiing area, using two distinct methods. The authors conclude that under the recorded production conditions around 40 % of the water used for MM snow is lost during the production, among which 10% are attributed to thermodynamic effects and 30 % to mechanical and/or technical effects (wind, topography...). I suggest accepting the paper but some moderate/major revisions are needed according to the listed specific comments.

**We thank Reviewer #3 for his/her thorough evaluation of our work. Some of the replies to his/her questions and issues are common to the replies to Reviewer #1 and Reviewer #2 and were not repeated.**

**Specific comments:**

**Done** In general the manuscript is well structured but requires a general overview by a an english mother tongue.

See comment **[1A]**

**Done** Moreover it should focus mainly on the main outcomes of the research, with a reduction of some sections that now are too wordy. In particular the abstract and the conclusions should be shortened.

See comment **[1D]** and **[1F]** and **[2A]**

**Specific comments are listed below:**

**Done** Pag 1 line 25: I think that snowmaking is not only related to the mitigation/adaptation of the effects of climate change but also to face the natural variability of snow conditions. Keep in mind the concept of natural variability as you reported in the abstract (Pag 1 line2).

We agree. Thank you for this comment.

Pag 2 line 5: See also: Signatures of Evaporation of Artificial Snow in the Alpine Lower Troposphere (SEASALT) Geophysical Research Abstracts, Vol. 10, EGU2008- A-11002, 2008 SRef-ID: 1607-7962/gra/EGU2008-A-11002 EGU General Assembly 2008

Pag 2 line 21: How did they explain such differences between fan and air water guns?

**[3A]** Olefs et al. (2010) only reported that "Such losses are estimated to be around 5%–15% for fan guns and 15%–40% for air–water guns (various snow gun producers 2009, personal communication).

Please refer to Olefs, M.; Fischer, A. & Lang, J. Boundary Conditions for Artificial Snow Production in the Austrian Alps. J. Appl. Meteor. Climat., 2010, {49}, 1096-1113

Pag 2 line 23: How the vegetation may influence the efficiency of MM snow?

**[3B]** The vegetation may capture the MM snow if close enough to the snowgun and favour the sublimation. Please refer to the reference in comment **[2I]** and:

Pomeroy, J., Parviainen, J., Hedstrom, N., Gray, D., 1998. Coupled modelling of forest snow interception and sublimation. Hydrol. Process. 12, 2317–2337. http://dx.doi.org/10.

**Done** Pag 2 line 24: Here the authors report as external factors wind, topography and vegetation. At pag 1 line 18 you reported human decisions and not vegetation. Please here and in the all text consider the same external factors.

Pag 2 line 30: Among the available data on the snow production why you don't consider the relative humidity in air?

**[3C]** The wet-bulb temperature Tw was used to characterize the meteorological conditions. Tw is based on the dry air temperature and the humidity of the air. This is the most employed variable by professionals to control the production of MM snow.

See references in comments **[3A]** and **[2I]** for more details

**Done** Pag 3 line 4: What do you mean with the term important? Do you refer to the number of skiers that ski along the slope?

If referring to line 6 of page 3, this is in term of number of skiers and also to allow to skiers to return back down to the village by skiing (a structural slope for skiers flows)

Pag 9 line 10: At what depth did you measure the snow density? Do you assume that the snow density was constant at the different depths in the snow piles? I think that the snow density could be higher at greater depths in the snow piles.

We measured the snow density at the surface of the snow pile and we assumed the density was constant at different depths.

We expect this assumption to be realistic since the MM snow initial density was very high (over 400 kg/m3) and since our observations were realized within hours after the production stopped (usually immediately after). This would have been highly different with natural snow density conditions.

What are the main results from the application of the PICO coring auger (Pag 9 line 13).

Results are reported in Table 3. Locations of the density measurements can be visualized in figure 4

Pag 11 line 3: Do you assume a higher melting rate due to the reduction of albedo caused by the deposition of dry particles?

Yes we do to match the observed melting rate in natural snow conditions.

In a similar way to: Dumont, M., Durand, Y., Arnaud, Y., and Six, D.: Variational assimilation of albedo in a snowpack model and reconstruction of the spatial mass-balance distribution of an alpine glacier, J. Glaciol., 58(207), 151 – 164, doi:10.3189/2012JoG11J163, 2012.

Pag 20 line 10: I think that the authors should provide more details about the methodology elaborated by Eisel (1988) to calculate sublimation/evaporation. Since this is the main source

of potential mass loss beside mechanical effects, this subject shoud be better clarified in the whole paper.
See comment **[1C]**

Pag 21 line 31: Among the ideal conditions I would add also the low water content in air.
See comment **[3C]**

Pag 23 line 6: Do you mean that in December you had an extreme meteorological event?
We did not report any extreme event. We do not understand this comment.

Please take into account that December 2015 was exceptionally mild with little snow. How this extreme meteorological event affect the snowmaking in the study area? You reported this point in the conclusions but not in the Discussion section.
December 2015 did not show an extreme event in the study area. Similar conditions have already occurred (e.g. December 2001) and will occur again (e.g. December 2016)

**Technical corrections:**
**Done**  Pag 3 line 4: add . after a.s.l.